

# The Stochastic Ice-Sheet and Sea-Level System Model v1.0 (StISSM v1.0)

Vincent Verjans[1], Alexander A. Robel[1], Helene Seroussi[2], Lizz Ultee[3], and Andrew F. Thompson[4]

[1]School of Earth and Atmospheric Sciences, Georgia Institute of Technology, Atlanta, GA, USA
[2]Thayer School of Engineering, Dartmouth College, Hanover, NH, USA
[3]Department of Earth and Climate Sciences, Middlebury College, Middlebury, VT, USA
[4]Environmental Science and Engineering, California Institute of Technology, Pasadena, CA 91125, USA

**Correspondence:** Vincent Verjans (vverjans3@gatech.edu)

**Abstract.** We introduce the first version of the Stochastic Ice-sheet and Sea-level System Model (StISSM v1.0), which adds stochastic parameterizations within a state-of-the-art large-scale ice sheet model. In StISSM v1.0, stochastic parameterizations target climatic fields with internal variability, as well as glaciological processes exhibiting variability that cannot be resolved at the spatiotemporal resolution of ice sheet models: calving and subglacial hydrology. Because both climate and unresolved
glaciological processes include internal variability, stochastic parameterizations allow StISSM v1.0 to account for the impacts of their high-frequency variability on ice dynamics, and on the long-term evolution of modeled glaciers and ice sheets. StISSM v1.0 additionally includes statistical models to represent surface mass balance and oceanic forcing as autoregressive processes. Such models, once appropriately calibrated, allow users to sample irreducible uncertainty in climate prediction without the need of computationally expensive ensembles from climate models. When combined together, these novel features of StISSM v1.0
enable quantification of irreducible uncertainty in ice sheet model simulations, and of ice sheet sensitivity to noisy forcings. We detail the implementation strategy of StISSM v1.0, evaluate its capabilities in idealized model experiments, demonstrate its applicability at the scale of a Greenland ice sheet simulation, and highlight priorities for future developments. Results from our test experiments demonstrate the complexity of ice sheet response to variability, such as asymmetric and/or non-zero mean responses to symmetric, zero-mean imposed variability. They also show differing levels of projection uncertainty for stochastic
variability in different processes. These features are in line with results from stochastic experiments in climate and ocean models, as well as with the theoretical expected behavior of noise-forced non-linear systems.

## 1  Introduction

Process-based numerical ice sheet models (ISMs) are the principal tool for projections of future mass balance of the Greenland and Antarctic ice sheets, and their future contribution to sea-level rise. They simulate gravity-driven ice flow, given some cli-
matic forcing and boundary conditions at the surface, basal, and lateral boundaries. In the past decade, a number of physically-based ISMs aimed at simulating large-scale ice sheet dynamics and projecting future sea-level rise have been developed and/or substantially improved (e.g., Larour et al., 2012; Cornford et al., 2013; Gagliardini et al., 2013; Pattyn, 2017; Hoffman et al., 2018; Lipscomb et al., 2019; Berends et al., 2021). Recently, important advances have been made in including key physical





processes (Bondzio et al., 2016), data assimilation (Goldberg and Sergienko, 2011), adaptive grid refinement (dos Santos et al.,
2019), and coupling with external climatic models (Gladstone et al., 2021), among other developments. In this context, there
has been a growing interest in performing model intercomparison experiments, in which different models simulate ice sheet
evolution over a given set of climatic projections. Such studies capture a range of possible future behaviors of the Greenland
and Antarctic ice sheets, depending on the level of global warming, and quantify the uncertainty associated with discrepancies
between ISMs. These recent efforts have culminated in the Ice Sheet Model Intercomparison Project for CMIP6 (ISMIP6)
(Nowicki et al., 2020; Goelzer et al., 2020a; Seroussi et al., 2020), which provides the basis for estimates of future sea-level
rise from the ice sheets in the IPCC AR6 report (Masson-Delmotte et al., 2021).

Different ISM responses to climatic forcing stem from a variety of ISM differences related to inclusion of physical processes,
approximation of ice flow equations, model resolution, initial conditions and geometry, numerical methods, and parameteriza-
tion of poorly constrained processes (e.g., Goelzer et al., 2018; Seroussi et al., 2019). The latter source is arguably the most
investigated source of uncertainty in ISMs, because it pertains to influential parameters of which direct observations are often
non-existent at ice sheet scale (Pollard and DeConto, 2012). Well-studied examples include the basal sliding coefficient and
the ice viscosity factor; these parametric uncertainties in ice sliding and viscosity are the focus of numerous inverse technique
applications (e.g., Morlighem et al., 2010; Petra et al., 2012; Pollard and DeConto, 2012; Perego et al., 2014), as well as
uncertainty quantification studies (e.g., Schlegel et al., 2018; Aschwanden al., 2019; Bulthuis et al., 2019).

In addition to discrepancies between ISMs, another large uncertainty in ISM projections is attributable to future atmospheric
and oceanic forcings (Nowicki and Seroussi, 2018; Pattyn et al., 2018). Realizations of possible forcings are provided by Global
Climate Models (GCMs), which can be further downscaled with Regional Climate Models (RCMs). However, running GCMs
or RCMs requires substantial computational resources, prior to running the ISM. This limits the number of available climatic
forcing scenarios that resolve the features required for ISM forcing in model ensemble studies. For example, for Antarctic
simulations under a high-emission scenario, ISMIP6 used six climatic forcing scenarios, each generated from a different GCM.
However, small round-off level differences in GCM initial conditions can, by themselves, yield a large spread in projected
climatic fields (Kay et al., 2015; Maher et al., 2019). This is often referred to as internal climate variability, aleatoric climatic
uncertainty, or irreducible climatic uncertainty. As a consequence, for any choice, configuration, and parameterization of GCMs
and ISMs, internal climate variability is an additional and unavoidable source of uncertainty. Internal climate variability is not
only a model feature, but it has also been extensively validated through observational climatic records (Mitchell, 1976; Chylek
et al., 2012; McKinnon et al., 2017). It has been demonstrated that the response of ice sheet dynamics to climatic forcing is
complex and non-linear (Huybrechts et al., 2011; Goelzer et al., 2013; Fyke et al., 2018; Robel et al., 2019), highlighting the
importance of understanding and quantifying ice sheet sensitivity to internal climate variability (Christian et al., 2022).

Furthermore, some glaciological processes exhibit variability on small spatiotemporal scales, and thus cannot be resolved in
current ISMs. Examples of such processes include iceberg calving, ice fracturing, and hydrology (Bassis, 2011; Hewitt, 2013;
Albrecht and Levermann, 2014; Kingslake, 2015). Parameterizing such small scale processes as deterministic forcings on the
ice sheet dynamic evolution involves simplifying assumptions, which often neglect internal variability of these processes. A
more accurate representation of these processes requires the inclusion of variability, rather than a constant forcing or parameter





value. Stochastic parameterization has been successfully used in ocean models and climate models for the last several decades
to address similar limitations, in subgrid-scale mixing processes for example (Hasselmann, 1976; Farrell and Ioannou, 1995;
Porta Mana and Zanna, 2014; Berner et al., 2017).

Studies with simple idealized models have demonstrated that mountain glaciers and marine-terminating glaciers are sensi-
tive to internal variability within the glacier and the climate systems (Oerlemans, 2000; Roe and Baker, 2014; Robel et al.,
2014, 2018; Christian et al., 2020, 2022). The magnitude of variability not only affects the magnitude of fluctuations in glacier
length and volume, but also glacier mean state, known as noise-induced drift (Hindmarsh and Le Meur, 2001; Mikkelsen et
al., 2018; Robel et al., 2018). Glaciers can also exhibit long-term fluctuations even when forced exclusively by inter-annual
climatic fluctuations, due to their long memory of past forcing (Roe and O'Neal, 2009; Robel et al., 2018). Mantelli et al.
(2016) showed that natural climate variability can push ice streams away from a state of stable behavior, as well as lead to
stable behaviors that do not exist in the absence of variability. Such theoretical studies have prompted interest in ISM simula-
tions accounting for climate variability. Consequently, other studies have specifically demonstrated asymmetric and non-linear
responses of West-Antarctic glaciers to ocean forcing variability (Snow et al., 2017; Hoffman et al., 2019; Robel et al., 2019).
Their results show that the glacier response to internal variability is complex, and modulated by ice dynamics, geometric glacier
configuration and statistics of the variability imposed.

Tsai et al. (2017, 2020) used large-scale ISM simulations to evaluate the impact of internal climate variability on Greenland-
and Antarctic-wide projections. Their approach used direct forcings from 40 to 50 members from coarse GCM ensemble
runs, and showed that the spread in ice sheet response due to internal climatic variability amounts to a significant fraction
of the mean response. Nonetheless, this approach faces some limitations. Firstly, GCMs produce coarse-resolution outputs
(~100 km grid scale, monthly steps) and cannot resolve processes with strong spatial gradients. Secondly, their outputs do
not have a one-to-one correspondence to inputs required for ISMs and thus require assumptions for such conversions. These
limitations can be addressed by forcing a RCM at its boundaries with the climatic fields from a GCM in order to dynamically
interpolate processes at a higher resolution, a method called dynamical downscaling. However, dynamically downscaling each
GCM ensemble member is computationally impractical. Finally, climate model simulations are generally not coupled to ISMs,
which neglects possible impacts from ice-sheet changes on the climate system, such as surface elevation changes and modified
ice discharge into the ocean.

In this study, we describe the first large-scale stochastic ISM, the Stochastic Ice-sheet and Sea-level System Model (StISSM
v1.0). StISSM v1.0 adds stochastic capabilities to the Ice-sheet and Sea-level System Model (ISSM, Larour et al. 2012). The
stochastic parameterizations target two different sources of variability. First, the variability internal to the ocean-atmosphere
system, but external to ice sheets. Second, inherent variability of some ice sheet processes, such as calving and subglacial hy-
drology. We refer to this collection of processes as "forcings with internal variability". The goal is to provide a straightforward
and efficient tool to the ice sheet modeling community for assessing irreducible uncertainty in ice sheet projections. StISSM
v1.0 has several advantages over more ad-hoc approaches to forcing models with variable fields: it gives the user the choice
of which processes exhibit stochasticity, it does not require direct forcing from ensemble runs of GCMs, it allows for different
spatiotemporal autocorrelations and correlations between processes, and it exploits parallelization to efficiently facilitate large



ensemble simulations. Statistics that determine the magnitude and the spatiotemporal dimensions of variability should be pro-
vided by users, and thus constrained from theory, observations or other model simulations (Bassis, 2011; Chylek et al., 2012;
Castruccio et al., 2014; Christensen, 2020; Hu and Castruccio, 2021). We underline that the purpose of StISSM v1.0 is not to
fully explore parametric uncertainty in ice sheet modeling, which is a separate task, and the subject of much ongoing research
(e.g., Schlegel et al., 2018; Aschwanden al., 2019; Bulthuis et al., 2022). Instead, it is the first integrated computational tool
that focuses on the impacts of internal variability on large-scale ISM simulations, with an emphasis on usability.

## 2  Methods

Stochastic capabilities are implemented within the core of the source code of ISSM. We refer readers to Larour et al. (2012)
for a general description of ISSM. Usage of stochasticity is optional: if turned off, ISSM simulations are fully deterministic.

### 2.1  Stochastic fields

Stochastic variability can be applied to a number of variables in ISSM, independently or with intervariable correlation. The
stochastic variables implemented for v1.0 of StISSM encompass both climatic forcings, and unresolved glaciological pro-
cesses: Surface Mass Balance (SMB), ocean forcing, calving and subglacial water pressure. These variables were prioritized
for the implementation of stochasticity because they are known to be subject to internal climate variability (SMB and ocean
forcing) and/or they reflect the impact of unresolved small scale processes (calving and subglacial water pressure) (Ribergaard
et al., 2008; Bassis, 2011; Hewitt, 2013; Fettweis et al., 2020). Our modeling framework ensures that, in the future, stochas-
tic variability can easily be implemented for other variables and parameters of ISSM. In StISSM v1.0, the model schemes
that parameterize these variables selected for stochasticity are: prescribed SMB, autoregressive SMB, prescribed calving rate,
prescribed floating ice melt rate, depth-dependent parameterization of floating ice melt, autoregressive depth-dependent pa-
rameterization of floating ice melt, autoregressive thermal forcing for terminus melting, and subglacial water pressure. We give
here a brief description of the different stochastic variables implemented in StISSM v1.0 and their parameterization.

The notion of "prescribed" means that the values of a variable are explicitly provided by the user, as opposed to calculated
within ISSM. They can be prescribed as either single values or varying in space and/or time. Turning on stochasticity for such
variables in StISSM v1.0 implies that Gaussian noise is added to the prescribed values at a user-defined temporal frequency in
the simulation. Gaussian noise has zero mean by definition. In StISSM v1.0, a generic variable $y$ with a prescribed mean value
$\overline{y}$ is represented as:

$$
\begin{cases}
y(t_k) = \overline{y} + \epsilon_y(t_k) \\
\epsilon_y \sim N\left(0, \sigma_y^2\right)
\end{cases}, \tag{1}
$$

where $t$ represents time, $t_k$ denotes any model time step, and $\sigma_y$ is the standard deviation of the Gaussian noise applied to $y$,
which must be provided by the user.





A common depth-dependent parameterization of floating ice melt rate uses a piecewise-linear function in depth (e.g., Favier et al., 2014; Seroussi et al., 2014):

$$
m_{fl}(z) = \begin{cases} m_{fl,up}, & \text{if } z \geq z_{up} \\ m_{fl,up} + \frac{z - z_{up}}{z_{dp} - z_{up}}\left(m_{fl,dp} - m_{fl,up}\right), & \text{if } z_{up} > z > z_{dp} \\ m_{fl,dp}, & \text{if } z \leq z_{dp} \end{cases}
\tag{2}
$$

where $m_{fl}(z)$ is the melt rate [m ice equivalent (m ice eq.) yr$^{-1}$] at a given vertical level $z$ [m], constrained by $m_{fl,up}$ at $z_{up}$ and $m_{fl,dp}$ at $z_{dp}$. In this parameterization, noise is added to the mean value of the floating ice melt rate at depth, $\overline{m}_{fl,dp}$. As such, $m_{fl,dp}$ follows Eq. (1), where $m_{fl,dp}$ is substituted for $y$. The rationale for stochastic variability in $m_{fl,dp}$ is, for example, to represent variability in deep water temperature below Antarctic ice shelves, which is known to have a strong impact on ice dynamics (Jenkins et al., 2018).

The deterministic subglacial water pressure, $\overline{p}_w$, can be calculated in different ways in ISSM. First, it can be forced to 0 over the entire domain ($\overline{p}_w = 0$), assuming the absence of subglacial water. Second, it can be computed as

$$
\overline{p}_w = \rho_w g(z_{sl} - z_b),
\tag{3}
$$

where $\rho_w$ is the density of water [kg m$^3$], $g$ is gravity [m s$^{-2}$], $z_{sl}$ is the sea-level elevation [m], and $z_b$ is the elevation of the base of the ice column [m]. Third, it can be computed as in Eq. (3) but set to 0 if negative. In each of these three cases, applying stochasticity adds Gaussian noise to the mean value $\overline{p}_w$, which corresponds to using Eq. (1) with $p_w$ substituted for $y$. The water pressure is subsequently used in the model to calculate the effective pressure $N_{eff}$:

$$
N_{eff} = \rho_i g H - p_w,
\tag{4}
$$

where $\rho_i$ is the density of solid ice, and $H$ is the thickness of the ice column. $N_{eff}$ is used in various parameterizations of the basal sliding speed available in ISSM (Brondex et al., 2019), allowing $\epsilon_{p_w}(t_k)$ to influence ice dynamics.

Thermal forcing, $TF$ [K], quantifies the excess ocean temperature with respect to the freezing point of water at the interface between the front of grounded outlet glaciers and the ocean. $TF$ thus enters in the parameterization of melt rates at the terminus of outlet glaciers, $m_{trm}$. In ISSM, $m_{trm}$ follows the formulation of Rignot et al. (2016):

$$
m_{trm} = (A h_w q_{sg}^{\alpha} + B) TF^{\beta},
\tag{5}
$$

where $q_{sg}$ is the subglacial water flux [m d$^{-1}$], $h$ is water depth [m], and $A$, $B$, $\alpha$, and $\beta$ are calibration parameters. Equation (5) applies to the front of grounded outlet glaciers rather than under ice shelves, thus more representative of Greenland than Antarctic conditions. Variability of ocean temperatures around Greenland has a large impact on outlet glacier dynamics (e.g., Straneo and Heimbach, 2013; Wood et al., 2021), motivating our choice to prioritize stochastic variability in $TF$. Thus, in StISSM v1.0, $TF$ can be modeled as an autoregressive process, as detailed in Section 2.3.





## 2.2 Numerics and spatiotemporality of stochasticity

Allowing all the $\epsilon_t$ terms introduced in Section 2.1 to vary in time and space implies that the spatial dimensions of stochasticity and the stochastic time step should be specified; these concepts are explained in this section.

The stochastic time step corresponds to the temporal frequency at which new noise terms for the stochastic variables are computed. The only restriction on the choice of the stochastic time step is that it cannot be smaller than the main time step of the numerical model simulation, i.e., the time step used by ISSM. It is important to specify the stochastic time step separately from the simulation time step, because the variability of a time series depends not only on the amplitude of the noise imposed, but also on the temporal frequency at which the noise is imposed. As such, if the stochastic time step was simply set equal to the simulation time step, changing the simulation time step would modify the variability imposed by the forcings to the ice sheet. When the stochastic time step is larger than the numerical model time step, then the noise term is not changed on every model time step. Integrating the noise in this fashion, and requiring the provided noise parameters to be self-consistent with the stochastic time step, means that the ice sheet responds to a forcing with characteristics of variability unaltered by other numerical considerations. Thus, the noisy forcing frequency and amplitude remain independent of the numerical model time stepping scheme, and the latter does not influence the sensitivity of the ice sheet to stochastic variability.

The spatial dimensions of stochasticity account for the number of sub-domains of the computational domain that share the same noise terms. The stochastic fluctuations are uniformly applied in each separate sub-domain. For example, a domain could be separated into individual glacier catchments. The number of sub-domains is prescribed during the parameterization of the model, and can be as large as the number of mesh elements in the domain.

StISSM v1.0 computes all noise terms according to a Gaussian distribution. The Gaussian noise can have different correlation features in space, in time, and between variables. While future work will focus on better constraining statistical distributions of variability in the processes of interest, many geophysical processes fluctuate with a Gaussian distribution when integrated over time (e.g., Hasselmann, 1976). The distribution of the noise is multivariate if stochasticity is applied to several variables, or if a variable has a spatial dimensionality greater than one:

$$\epsilon(t_k) \sim N(\mathbf{0}, \Sigma), \tag{6}$$

where $\epsilon_t$ is a vector of size of the total stochastic dimensionality, $d_{tot}^{st}$, and $\Sigma$ is the global covariance matrix of size $d_{tot}^{st} \times d_{tot}^{st}$. $d_{tot}^{st}$ is the sum of the number of stochastic variables times their respective spatial dimension. In this way, covariance entries can be prescribed between all the sub-domains and between any variable, thus including the different climatic forcing variables and the glaciological processes. The covariance matrix prescribed must be a valid covariance matrix, i.e., positive semi-definite with the variances $\sigma^2$ along the diagonal. It is the responsibility of the user to prescribe covariance entries that suit their interest, which can be challenging, depending on the specifics of the problem investigated (e.g., Hu and Castruccio, 2021). Examples of relevant covariances include: between SMB and subglacial water pressure, between floating ice melt rates and calving, between *TF* in different sub-domains, between SMB and *TF*, etc. The simplest choice is to set all non-diagonal entries to 0, which corresponds to statistical independence between all variables and all sub-domains.



From the stochastic time step, each simulation time step is determined as being a stochastic model step or not. At a stochastic model step, new $\epsilon_t$ terms are generated from Eq. (6). Otherwise, $\epsilon_t$ values from the previous step are re-used. At a stochastic
model step, all $\epsilon_t$ terms are computed simultaneously, and before the solvers of the ISSM governing equations. Because $\epsilon_t$ terms are specific to sub-domains, an $\epsilon_t$ term computed for a given sub-domain is subsequently assigned to the nodal points of all the elements belonging to the sub-domain. Stochasticity has been implemented in the most general way possible, such that developing stochasticity for a new variable would only require reproducing the code from another variable, with minimal adaptations needed for variable names and for potential specificities of the new variable. Moreover, the stochastic noise
generation is mostly implemented in a separate module, thus causing minimal interference to developments of any other aspect of ISSM. The random number generator implemented in ISSM is the commonly used linear congruential generator, which is a recursive algorithm with the advantages of being fast and easy to implement (Knuth, 1998). For the sake of reproducibilty of results or troubleshooting, a randomness flag can optionally be set to false during the configuration of a simulation of StISSM v1.0. In this case, each random number generation uses a seed that is a deterministic function of the time step.

## 2.3    Autoregressive schemes

In an autoregressive (AR) process, the variable of interest evolves in time, and depends linearly on its own previous values. AR models are a powerful tool in climatic time series analysis because they are discretized versions of differential equations. They capture important features such as seasonality and characteristic time scales of geophysical processes, and they have been shown to characterize many complex climatic variables (Hasselmann, 1988; von Storch and Zwiers, 1999). Long-term records
of accumulation, and atmospheric and oceanic temperatures in Greenland and Antarctica are commonly represented by AR processes (e.g., Roe and Steig, 2004; Thomas et al., 2009; Mikkelsen et al., 2018; Rosier et al., 2021). For all these reasons, we have implemented AR capabilities in StISSM v1.0, but more complex time series models can be implemented in future developments. A general AR model of order $p$ for an autocorrelated variable $\eta$ at a time step $t_k$ is given as:

$$\eta(t_k) = \beta_0 + \beta_1 t + \sum_{i=1}^{p} \varphi_i \eta(t_{k-i}) + \epsilon_\eta(t_k), \tag{7}$$

where $\epsilon_\eta$ is a Gaussian noise term uncorrelated in time, $\varphi_i$ are lag coefficients, $\beta_0$ is an intercept and $\beta_1$ is a trend.

We have implemented AR options in StISSM v1.0 for the three climate-related variables mentioned in Section 2.1: (i) SMB, (ii) deep-water melt rate, $m_{fl,dp}$, when using the depth-dependent parameterization Eq. (2), and (iii) thermal forcing for terminus melting, *TF* (Eq. (5)). Thus, any of these variables can be computed with an AR model, following Eq. (7). AR capabilities can be turned on for a single variable or for multiple variables. The order $p$ and all the coefficients $\beta_0$, $\beta_1$, and $\varphi_i$ are prescribed
by the user, and are specific to the variable chosen. The latter parameters are all fixed in time, but can vary in space as detailed below.

All the variables computed via an AR model have their specific spatial dimensions and temporal setting. The spatial dimensions work in the same way as for the generic stochasticity (Section 2.2). The coefficients $\beta_0$, $\beta_1$, and $\varphi_i$ are thus specific to each sub-domain, as are the $\epsilon_\eta$ terms computed in Eq. (7). The sub-domains of the AR variables need to be specified, and do
not need to be identical to the sub-domains of the other stochastic variables. Noise terms for Eq. (7) are generated for each



sub-domain of the AR variables. The global covariance matrix must include covariance terms between AR sub-domains and with the sub-domains of other stochastic variables. The AR time setting corresponds to $t$ in Eq. (7). A variable $\eta(t_k)$ is recomputed following Eq. (7) at the frequency determined by $t_k$, and remains constant in-between if the ISSM simulation time step is shorter than successive $t_k$ steps. The AR frequency cannot be higher than the stochastic frequency such that a new $\epsilon_\eta$ term
is available every time $\eta(t_k)$ is recomputed via Eq. (7). Throughout a simulation, StISSM v1.0 retains in memory only those terms needed to compute AR variables at the present simulation time step, the number of which depends on the order $p$ in Eq. (7).

The SMB AR scheme optionally allows for dynamic SMB-elevation feedback through prescribed altitudinal gradients of SMB. Such gradients, called SMB lapse rates, relate SMB values at individual mesh elements to varying elevation in space
and/or time, and are regularly applied for Greenland ice sheet simulations (Edwards et al., 2014; Goelzer et al., 2020b). Each sub-domain of the SMB AR scheme has its specific set of lapse rates and their corresponding elevation ranges in which they apply. There is no limit on the number of different lapse rates per sub-domain, as long as each lapse rate has a corresponding elevation range. The lapse rates serve to adjust SMB at individual mesh elements as a function of their elevation, with respect to the sub-domain reference SMB value calculated via Eq. (7). Furthermore, they allow for dynamic SMB feedbacks in time
as ice thickens or thins throughout a simulation.

### 2.4 Ensemble simulations

In order to sample the component of irreducible uncertainty due to internal and climate variability, StISSM v1.0 allows for ensemble runs of the ice sheet model with stochastic parameterizations activated. Each ensemble is characterized by a selection of stochastic variables, and a given configuration of the stochasticity (Sections 2.1, 2.2). The number of simulations, referred
to as ensemble members, is chosen by the user; each member is then characterized by a unique stochastic realization. All the simulations for the different members can be run in parallel, allowing for efficient simulations. The implementation of the ensemble simulations is straightforward, as illustrated in Algorithm 1. The procedure for launching simulations does depend on the system and scheduler being used. StISSM v1.0 includes examples of ensemble launchers for common task schedulers in computer clusters.

## 3 Model experiments


We perform three sets of experiments to test and demonstrate the new capabilities of StISSM v1.0. The first set simulates a marine-terminating glacier with geometry taken from the benchmark configuration of the Marine Ice Sheet Model Intercomparison Project (MISMIP+, as described in Asay-Davis et al. 2016). The second set simulates a quarter of an idealized circular ice sheet (IQIS) with a fast-flowing ice stream. The third simulates the Greenland ice sheet (GrIS). We detail the configuration
of the three demonstration experiments in this section.

The MISMIP+ configuration is a well-known and thoroughly tested configuration. While this is the first study applying stochasticity to MISMIP+, our results can be compared to prior studies with a large range of different ISMs (Asay-Davis



---

**Algorithm 1** Parallelization procedure for ensemble runs

---

    choose stochastic variables

    configure stochasticity

    $n \leftarrow$ number of ensemble members

    **for** $ii = 0$ to $n$ **do**

        $name_{ii} \leftarrow$ name of simulation $ii$

        launch simulation of $name_{ii}$

    **end for**

    [$n$ simulations run in parallel]

    **for** $ii = 0$ to $n$ **do**

        retrieve results of $name_{ii}$

    **end for**

---

et al., 2016; Seroussi and Morlighem, 2018; Cornford et al., 2020). The IQIS design resembles that of a real ice sheet, but with an idealized setup. This configuration allows us to investigate the role of different types of stochastic forcings, without complications from more realistic setups. Finally, the GrIS simulations demonstrate the applicability of StISSM v1.0 to a realistic model configuration on an entire ice sheet, which is a necessary consideration for our ice sheet modeling objectives.

### 3.1 MISMIP+

Our MISMIP+ experiment follows the description and parameterizations of the MISMIP+ Ice1r design in Asay-Davis et al. (2016) and Experiment 2 of Seroussi and Morlighem (2018). The domain spans 640 km in the x-direction and 80 km in the y-direction, with a fixed ice front at x = 640 km (Fig. 1). Along the centerline y = 40 km, the bed topography is 150 m below sea-level at x = 0 km, and progressively decreases in the x-direction, but with a bed bump between x = 390 and 506 km (Fig. 1). The spin-up run starts from a thin (10 to 100 m thickness) glacier over the entire domain, is fully deterministic, has a horizontal resolution of 1 km, and uses the Shallow-Shelf Approximation (MacAyeal, 1989). The glacier progressively builds up through a constant positive SMB set to 0.3 m ice eq. yr$^{-1}$, and uniform over the domain. For the melting rates applied to floating ice, we use the depth-dependent melting parameterization Eq. (2). We fix the values $m_{fl,up}$ = 0 m ice eq. yr$^{-1}$, $z_{up}$ = -50 m, $m_{fl,dp}$ = 1 m ice eq. yr$^{-1}$, and $z_{dp}$ = -500 m. Similarly to Seroussi and Morlighem (2018), we use a Weertman-type sliding law (Weertman, 1957):

$$\boldsymbol{\tau_b} = -C_W^2 ||\boldsymbol{u_b}||^{\frac{1}{m}-1} \boldsymbol{u_b}, \tag{8}$$

where $\boldsymbol{\tau_b}$ is the basal stress [Pa], $\boldsymbol{u_b}$ is the basal velocity [m yr$^{-1}$], $m$ = 3, and $C_W^2$ is a basal drag coefficient taken equal to 1.0×10$^4$ Pa m$^{-1/3}$ yr$^{1/3}$. A steady-state is reached after 19 000 years, at which point SMB is balanced by the melt of floating ice and ice flow out of the model domain. The time step used during the spin-up is 1/2 yr over the first 15 000 years, and refined to 1/4 yr over the last 4000 years. We use this steady-state as an initial state for transient experiments. Under such relatively low melt conditions, the initial state is characterized by a thick ice-shelf, buttressing and stabilizing the grounded part of the



glacier, and the grounding line along the center flowline is located within a bed trough at x = 397.0 km (Fig. 1). The relative
changes in ice mass and in grounded ice area over the last 1000 years of the spin-up are both $< 0.15\%$.

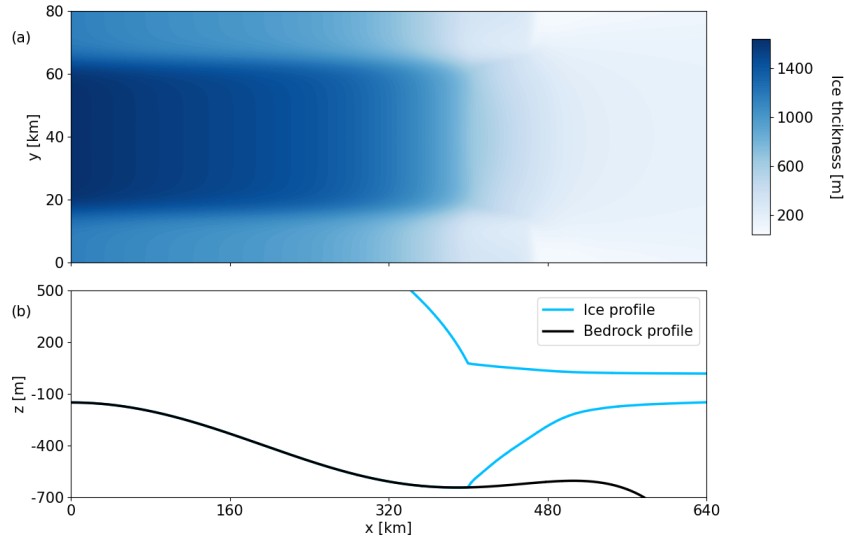

**Figure 1.** Initial steady-state configuration for the MISMIP+ experiments. (a) Ice thickness in (x,y) plane, (b) vertical profile along y = 40 km.

From this steady-state, we perform 5 ensembles of transient simulations. The transient experiments are performed over a period of 500 years, with a time step of 1/4 yr, and with the same spatial resolution of 1 km as in the spin-up run. Each ensemble consists of 500 individual simulations, referred to as members. Each ensemble member has the mean melt rate signal of $\overline{m}_{fl,dp} = 1$ m ice eq. yr$^{-1}$, but applies stochastic perturbations $\epsilon_{m_{fl,dp}}$ (Eqs. (1), (2), (7)). While $\overline{m}_{fl,dp}$ is equal to the forcing melt

rate in the spin-up, the stochastic perturbations will cause deviations of the glacier from its initial deterministic steady-state. Ensembles differ in the statistics of the stochastic perturbations, while members of a same ensemble only differ by their unique realization of random noise. We apply no trend in $m_{fl,dp}$, and use a stochastic time step of 1 year, i.e., annual variability in the stochastic perturbations. Through our five ensembles, we evaluate the sensitivity of the model configuration to persistence of ocean melt variability in time. All the ensembles represent $m_{fl,dp}$ as an order 1 (i.e., $p = 1$ in Eq. (7)) AR process, commonly

written AR(1). In an AR(1) model, the autocorrelation coefficient $\rho_1$ is equivalent to $\varphi_1$ in Eq. (7), and can be converted to a more intuitive characteristic timescale, $\tau$, via (e.g., Burke and Roe , 2014):

$$\tau = \frac{\Delta t}{1 - \rho_1},\qquad(9)$$

where $\Delta t$ is the time-discretization of the stochastic time series, here set to 1 year. We test a range of five values of $\tau$: 1, 2.5, 10, 20 and 50 years. Note that $\tau = 1$ year corresponds to annual uncorrelated white noise (equivalent to Eq. (1)). We denote these

five ensembles as $MISMIP_{\tau=1}$, $MISMIP_{\tau=2.5}$, $MISMIP_{\tau=10}$, $MISMIP_{\tau=20}$, and $MISMIP_{\tau=50}$, respectively. For all ensembles,




the standard deviation of the Gaussian noise term is set to 1/3 of the deterministic mean $\overline{m}_{fl,dp}$, $\sigma_{m_{fl,dp}}$ = 1/3 m ice eq. yr$^{-1}$:

$$\epsilon_{m_{fl,dp}} \sim N\left(0, \sigma^2_{m_{fl,dp}}\right)$$

for $MISMIP_{\tau=1,2.5,10,20,50}$ $\qquad\qquad\qquad\qquad\qquad\qquad\qquad$ (10)

In all the MISMIP+ transient experiments, the $\epsilon_{m_{fl,dp}}$ perturbations are uniformly applied over the entire model domain.

## 3.2 Idealized Quarter Ice Sheet

The Idealized Quarter Ice Sheet (IQIS) experiment is performed on a square domain of $750\times750$ km$^2$ and meant to represent a quarter of a circular marine ice sheet in an idealized configuration (Fig. 2). We impose Dirichlet conditions with values of 0 m/yr for the x-direction velocities at y = 0 m and y-direction velocities at x = 0 m to represent ice divides. By symmetry, the four quarters of the fully circular ice sheet would each be identical, thus we only simulate one quarter to save computational expense. The bed topography decreases linearly along the radius, from 10 m above sea-level at the origin (lower-left corner) to

20 m below sea-level at (x = 750 km, y = 750 km, upper-right corner), corresponding to a very gradual bed slope in the radial direction of $4 \times 10^{-5}$. The friction law used for parameterizing basal sliding is the Budd friction law (Budd et al., 1984):

$$\boldsymbol{\tau_b} = -C_B^2 \boldsymbol{u_b} N_{eff}, \qquad\qquad\qquad\qquad\qquad\qquad\qquad (11)$$

where $C_B^2$ is a basal drag coefficient set to 1600 m$^{-1}$ yr. We lower $C_B^2$ to 400 m$^{-1}$ yr on a 57 km wide strip starting 200 km away from the origin, simulating the presence of a fast-flowing ice stream (Fig. 2).

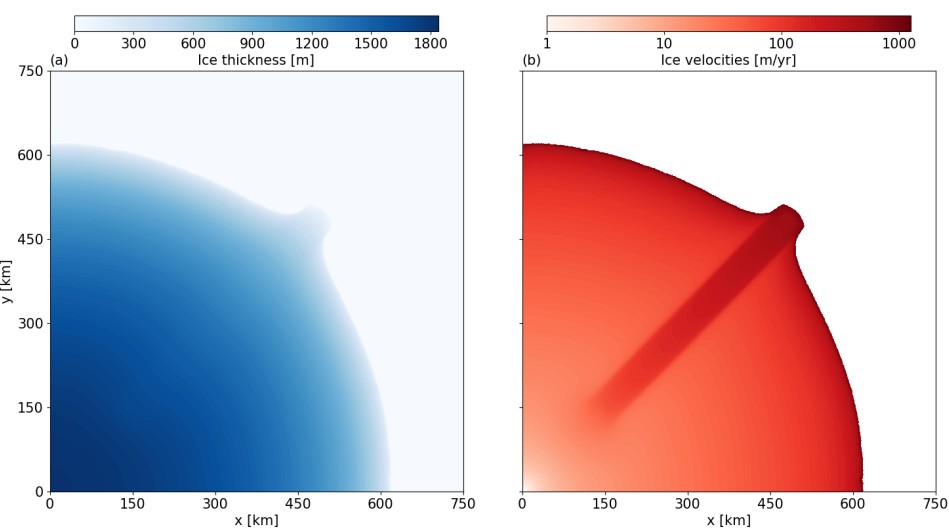

**Figure 2.** Initial steady-state configuration for the IQIS experiments. (a) Ice thickness, (b) ice velocities.

The deterministic steady-state of the ice sheet is constructed via a balance between constant positive SMB over the domain (0.4 m ice eq. yr$^{-1}$) and frontal ablation at the grounded ice-ocean front. We split the frontal ablation between a calving rate



flux, $c_{rate}$, and a frontal melt rate, $m_{trm}$ following Eq. (5) with $q_{sg} = 0$ m day$^{-1}$ and $TF = 4$ K. Note that $c_{rate}$ and $m_{trm}$ have the same effect: ice ablation at the terminus. The purpose of using these two terms is to allow for separate stochastic perturbations in calving and $TF$ during the transient experiments. We set $c_{rate}$ quadratically increasing along the radius, and this radial

dependence is enforced by a multiplicative factor, $c_{rate}^{fac}$, constant across the domain:

$$c_{rate} = c_{rate}^{fac} \left( \frac{rad}{750 \times 10^3} \right)^2, \tag{12}$$

where $rad$ denotes the radius [m], and $c_{rate}^{fac}$ is set to 690 m ice eq. yr$^{-1}$. This radial dependence allows the front to reach a stable position, where frontal ablation balances total SMB. The spin-up simulation is run for 79 000 years. In the final stage of the spin-up to steady-state (last 10 000 years), we use a horizontal resolution varying between 15 km in the interior and 800 m at

the front, a time step of 1/4 yr, and the Mono-Layer Higher-Order stress approximation. This stress approximation effectively captures the same longitudinal extension and vertically shearing flow as higher-order flow approximations, but without needing vertical model layers (dos Santos et al., 2022). The relative changes in total mass and area of the ice sheet over the last 1000 spin-up years are both $< 4 \times 10^{-3}\%$. The final steady-state ice thicknesses and velocities of IQIS are shown in Fig. (2). It has an ice-covered area of $\sim$31 000 km$^2$ and an ice mass of 350 715 Gt, equivalent to $\sim$1/5 of the area and $\sim$1/8 of the mass of the

Greenland ice sheet.

We use this steady-state as an initial state for transient ensemble experiments with stochasticity. We perform four sets of ensemble experiments (Table 1). Each of the four sets is characterized by activating stochastic fluctuations for specific forcings, while the mean forcings remain the same as in the spin-up. Thus, $\overline{\text{SMB}} = 0.4$ m ice eq. yr$^{-1}$, $c_{rate}^{fac} = 690$ m ice eq. yr$^{-1}$ (Eq. (12)), $q_{sg} = 0$ m day$^{-1}$, and $\overline{TF} = 4$ K (Eq. (5)). As such, our experiments quantify how fluctuations in different processes around a

given mean forcing can drive the IQIS away from its steady-state.

| | Stochastic variables | Mean forcing | $\sigma$ | $\tau$ [yr] | $r_{variables}$ | $r_{dom}$ |
|---|---|---|---|---|---|---|
| $IQIS_1$ | SMB | 0.4 m ice eq. yr$^{-1}$ | 0.13 m ice eq. yr$^{-1}$ | 1 | / | 0.6 |
| $IQIS_2$ | $c_{rate}$ | Eq. (12) | 164.5 m ice eq. yr$^{-1}$ | 1 | / | / |
| $IQIS_3$ | SMB, $c_{rate}$ | 0.4 m ice eq. yr$^{-1}$, Eq. (12) | 0.13 m ice eq. yr$^{-1}$, 164.5 m ice eq. yr$^{-1}$ | 1 | -0.6 | 0.6 |
| $IQIS_4$ | $TF$ | 4 K | 1.33 K | 10 | / | / |

**Table 1.** Configuration of the IQIS transient ensemble experiments. $\tau = 1$ yr corresponds to annual noise uncorrelated in time. $\sigma_{c_{rate}}$ is equal to 1/3 of the mean of Eq. (12) evaluated at the front location. Note that all $\sigma$ values are taken as 1/3 of the corresponding mean forcing. $r_{variables}$ is the noise correlation between variables ($\epsilon_{\text{SMB}}$ and $\epsilon_{c_{rate}}$ in the exterior sub-domain in $IQIS_3$). $r_{dom}$ is the correlation in $\epsilon_{\text{SMB}}$ between the interior ($rad \leq 300$ km) and exterior ($rad > 300$ km) parts of the domain.

The configuration of the IQIS transient experiments is detailed in Table 1. The experiments are designed to yield meaningful comparisons between ensembles with variability in different variables. $IQIS_1$ and $IQIS_2$ show the relative strength of SMB versus calving fluctuations on our experimental design. $IQIS_3$ shows the impact of combining the SMB and calving stochastic perturbations. Finally, $IQIS_4$ shows how decadal persistence of oceanic forcing at the ice front can impact ice sheet dynamics.



The ensemble experiments consist of 500 members, a simulation period of 500 years, and the spatial and temporal resolutions are kept identical to the final spin-up configuration. The stochastic time step is set to 1 yr, such that the fluctuations imposed have an annual sampling frequency. The first set, $IQIS_1$, applies annual white noise in SMB (see Eq. (1)). The standard deviation of the noise amplitude, $\sigma_{SMB}$ is taken as 1/3 of the mean (Table 1), comparable to the SMB relative inter-annual variability in Greenland (Fettweis et al., 2020). Here, we separate the domain into interior ($rad \leq 300$ km) and exterior ($rad > 300$ km) sub-domains. The correlation in $\epsilon_{SMB}$ between both parts is set to 0.6. The second set, $IQIS_2$, applies annual white noise in calving rates, $\epsilon_{c_{rate}}$ (see Eq. (1)). We choose the same 1/3 ratio of noise-to-signal as for SMB in $IQIS_1$ for better comparability. Because calving rates vary in space (Eq. (12)), we take $\sigma_{c_{rate}}$ as 1/3 of the mean calving rates at the ice front at the end of the spin-up phase (Table 1). The third set, $IQIS_3$, combines the stochastic forcings of $IQIS_1$ and $IQIS_2$ in a single ensemble experiment (Table 1). It uses a negative correlation between calving and SMB in the exterior sub-domain to represent, for example, the impact of increased surface melt on hydrofracturing (Benn et al., 2007). The last set of IQIS experiments, $IQIS_4$, models $TF$ (Eq. 5) as an AR(1) process (Eq. 7) without a trend and with the same mean as in the spin-up (4 K). We set $\sigma_{TF}$ to 1/3 of the mean signal for ease of comparisons with the other experiments, but here we apply a decadal noise persistence (Table 1). In Eq. (5), we use the parametrization of Rignot et al. (2016) and set $A = 3 \times 10^{-4}$, $B = 0.15$, $\alpha = 0.39$, and $\beta = 1.18$.

### 3.3 Greenland Ice Sheet

To demonstrate that StISSM v1.0 is readily applicable at the scale of ice sheet simulations, we simulate the evolution of the GrIS with stochastic SMB and ocean forcings. The configuration uses an initial state matched to observations, but is spun-up to reach a deterministic steady-state before launching the transient experiments with stochasticity applied. The initial state uses the bed topography, the ice thickness and the ice mask from BedMachine v4 (Morlighem et al., 2017), the ice velocity field from Joughin et al. (2017), the geothermal heat flux from Shapiro and Ritzwoller (2004), and surface temperatures from Ettema et al. (2009). We solve for a thermal steady-state in three dimensions, with 10 vertical layers, to compute and then depth-average an ice-rheology field following Cuffey and Paterson (2010). We invert for basal friction coefficients $C_B^2$ in the Budd sliding law (Eq. (11)). We use a linear regression of $C_B^2$ on bed elevation to extrapolate the field in areas covered by less than 500 m of ice to avoid spurious patterns in $C_B^2$.

After this initialization, we perform a deterministic spin-up in order to reach a GrIS configuration in steady-state. We emphasize that the purpose of our simulations is not to predict future ice mass balance of the GrIS, and we do not argue that the real GrIS is in steady-state; our goal with this spin-up is to have a steady baseline against which to compare a transient ensemble. We separate the GrIS in 19 different basins following an existing delineation (Zwally et al., 2012). The computation of SMB is basin-specific, and calibrated to the mean 1961-1990 modeled SMB field of RACMO2 (van Angelen et al., 2014). We fit piecewise linear functions of elevation with two breakpoints to the SMB data in order to derive three SMB lapse rates per basin, although only two lapse rates (i.e., a single breakpoint) are used if the fitting yields an unrealistic lapse rate over a narrow elevation range (Table 2). The reference SMB in each basin corresponds to the basin-specific piecewise linear function evaluated at the mean basin elevation (Table 2). Stochasticity is turned off during the spin-up, but SMB values of any mesh element do change in time, because they are adjusted as thickening and thinning patterns affect the elevation of the elements.





| Basin | Mean elevation [m] | Reference SMB ($SMB_{ref}$) [mm ice eq. yr$^{-1}$] | Elevation of SMB breakpoints [m] | SMB lapse rates [mm ice eq. yr$^{-1}$ m$^{-1}$] | Ocean $\overline{TF}$ [K] |
|---|---|---|---|---|---|
| 1 | 1795 | 179 | [858, 1202] | [0.86, 1.52, -0.028] | 2.0 |
| 2 | 1811 | 185 | [1213, 1594] | [0.86, 0.11, -0.12] | 2.0 |
| 3 | 1655 | 126 | [677, 1226] | [-0.048, 1.16, -0.052] | 2.0 |
| 4 | 1321 | 102 | [626, 1227] | [-0.056, 1.46, -0.009] | 2.0 |
| 5 | 2237 | 103 | [532, 1087] | [0.40, 1.64, 0.006] | 2.5 |
| 6 | 2159 | 125 | [1099, 1373] | [0.72, 1.02, -0.009] | 2.5 |
| 7 | 2484 | 231 | [1858, 2368] | [0.65, 0.27, -0.011] | 2.5 |
| 8 | 1749 | 560 | [1031, 1933] | [0.62, 0.21, -0.000] | 2.5 |
| 9 | 2461 | 691 | [665, 1428] | [1.01, - 0.14, -0.63] | 2.5 |
| 10 | 2335 | 756 | [1170] | [1.46, -0.66] | 6.5 |
| 11 | 1970 | 1498 | [1112, 2487] | [1.51, -1.78, -0.51] | 6.5 |
| 12 | 2150 | 1934 | [1577, 2116] | [0.619, -0.76, -2.05] | 6.5 |
| 13 | 1870 | 1253 | [1035, 2047] | [0.43, 0.66, -1.09] | 6.5 |
| 14 | 2117 | 801 | [1657, 2100] | [1.57, 0.40, -0.55] | 6.5 |
| 15 | 1900 | 483 | [1901] | [1.81, -0.063] | 6.5 |
| 16 | 2471 | 461 | [1640, 2586] | [1.78, -0.024, -0.36] | 4.5 |
| 17 | 2404 | 458 | [1605] | [1.58, -0.24] | 4.5 |
| 18 | 2197 | 371 | [1057, 1418] | [0.82, 0.28, -0.21] | 3.5 |
| 19 | 1269 | 518 | [1340] | [0.35, -0.44] | 3.5 |

**Table 2.** Climatic forcing in each basin for the GrIS simulations. Basin numbers correspond to the delineation in Zwally et al. (2012), shown in Fig. (3). Piecewise-linear functions of SMB to elevation are fitted to the RACMO2 SMB in each basin. The fit uses two breakpoints to derive three lapse rates. If the fit results in one of the lapse rate values being unrealistic and applying over a narrow elevation range, we use a fit with a single breakpoint and two lapse rates. The reference SMB corresponds to the piecewise linear function evaluated at the mean elevation. Ocean $\overline{TF}$ is the $TF$ value used in Eq. (5) to model frontal melt during the second phase of the spin-up at outlet glaciers where we parameterize ocean melt.

The spin-up itself is separated in two different phases, both of them using a weekly time step and the 2-dimensional Shallow-

Shelf Approximation. In the first phase, we fix the ice sheet margin positions, we implement free-flow boundary conditions at the ice margins, and we use a spatial resolution ranging from 25 km in the slowest flowing areas to 2 km in the fastest flowing areas. During this first spin-up phase, the modeled ice sheet adjusts to the SMB field until it reaches a steady-state. Steady-state of this first phase requires a dynamic equilibrium between ice flow and the SMB field, and thus takes 30 000 years.

In the second phase of the spin-up, we allow for moving margins at 11 of the major outlet glaciers of the GrIS, where we

parameterize ocean melt (Fig. 3). Resolving these 11 outlet glaciers and migration of their termini requires finer meshing close to their termini. Frontal ablation at the ice-ocean boundaries is applied through a combination of melting at the termini (Eq.





(5)) and calving. Values of $\overline{TF}$ are specific to each basin and taken as the approximate 1990-2018 average values reported by Wood et al. (2021) (Table 2). Calving values are prescribed as constant for each of the 11 glaciers, and taken to minimize the departure from the steady-state reached in the first phase (Table 3). We refine the mesh resolution at the 11 outlet glaciers 370 down to 800 m, while keeping the fixed front and free-flow boundary conditions for the other outlet glaciers. We select only 11 moving glaciers to avoid the computational load associated with refined horizontal resolution at all outlet glaciers. Our simulation set-up represents the impact of ocean forcing on the GrIS reasonably well, since ice discharge is dominated by a small (<20) number of glaciers (Enderlin et al., 2014).

| Glacier | I | II | III | IV | V | VI | VII | VIII | IX | X | XI |
|---|---|---|---|---|---|---|---|---|---|---|---|
| Calving rate [m ice eq. yr$^{-1}$] | 240 | 1100 | 5 | 5 | 4400 | 2050 | 2100 | 1000 | 600 | 2030 | 0 |

**Table 3.** Calving rates applied at the 11 outlet glaciers where terminus migration is simulated and ocean melt parameterized. Each Roman numeral is associated with a corresponding glacier, shown in Fig. (3). Total frontal ablation results from both calving and melt at the terminus, which is governed by basin-specific $\overline{TF}$ values given in Table 2

Comparing the simulated GrIS state at end of the spin-up to observations, the total ice mass and ice-covered area are ∼11 800 375 Gt (0.4%) and ∼71 000 km$^2$ (3.5%) lower, respectively. The main differences are an interior thickening in the South, and an inland retreat in the North (see Appendix A). Velocity magnitudes and patterns remain consistent with observed velocities (Fig. 3). The five simulated outlet glaciers in the North are retreated compared to observations, except for glacier II (Petermann). Over the last 200 years of spin-up, the changes in total ice mass and ice-covered area are +24.9 Gt (+0.001%) and -316 km$^2$ (-0.016%). More details concerning the GrIS steady-state are given in Appendix A.

We use the GrIS final steady-state as an initial state for our transient experiments with stochasticity turned on, which will cause deviations from the steady-state. We perform a single transient ensemble of 200 members over 500 years, with a stochastic time step set to 1 year, thus representing annual fluctuations. While this initial state is very close to a steady-state, we still perform a deterministic control run of 500 years, to quantify the minimal amount of deterministic model drift. In the stochastic transient ensemble, we apply stochastic fluctuations in the climate forcing fields SMB and $TF$. We represent both of these forc-385 ings as AR(1) processes (Eq. (7)). The means of the forcings are kept equal to the constant values applied during the spin-up (Table 2), and we do not impose any trend. We set $\tau$ = 3 yr for SMB (following Mikkelsen et al., 2018) and $\tau$ = 10 yr for $TF$. Each basin represents an individual spatial dimension for the covariance matrix (Fig. 3). Since our simulation set-up uses 19 basins and two variables with stochastic variability (SMB and $TF$), the covariance matrix is of dimensions $38 \times 38$. The standard deviations of the noise terms are all set to 1/3 of the mean forcing:

$$\begin{cases} \sigma_{\text{SMB},b} = \frac{1}{3}\text{SMB}_{ref,b} & \text{for } b = 1,...,19 \\ \sigma_{TF,b} = \frac{1}{3}\overline{TF}_b & \text{for } b = 1,...,19 \end{cases}, \qquad (13)$$

where the basin-specific reference SMB, SMB$_{ref,b}$, and $\overline{TF}_b$ forcings are given in Table 2. To prescribe all the correlations involving $\epsilon_{\text{SMB},b}$ and $\epsilon_{TF,b}$, and between all basins, we use simple and reasonably realistic assumptions: (i) a variable is more





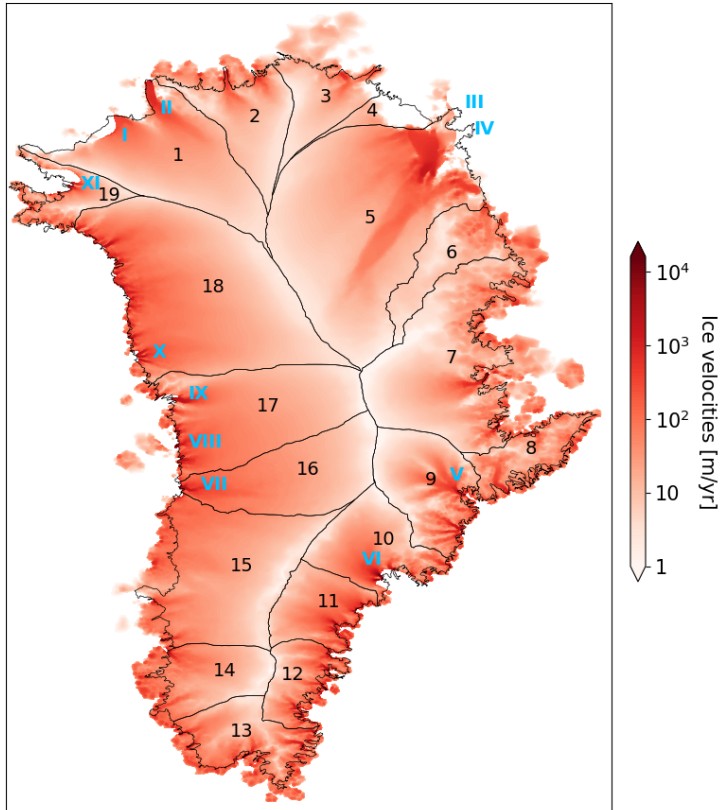

**Figure 3.** Steady-state ice velocities at the end of the second phase of the GrIS spin-up. Arabic numerals (black) show the individual basins. Roman numerals (cyan) show the outlet glaciers where ice front movement is simulated and ocean melt parameterized.

strongly correlated with values in the neighboring basins than in the non-neighboring basins, (ii) $\epsilon_{\mathrm{SMB}}$ and $\epsilon_{TF}$ are negatively correlated, and (iii) between different basins, $\epsilon_{\mathrm{SMB}}$ or $\epsilon_{TF}$ variables are more strongly correlated with themselves than with the

other variable. Under the constraint of using a positive semi-definite covariance matrix, all correlation absolute values range between 0.4 and 0.6 (see Appendix B for details). In the transient simulations, SMB lapse rates (Table 2), calving rates (Table 3), the weekly model time step, and the spatial resolution are kept identical to the second phase of the spin-up.

## 4 Results

In this section, we analyze the results of the transient MISMIP+, IQIS, and GrIS experiments in terms of total ice mass [Gt]
evolution. While our analyses focus on a variable summed over the entire domain, we note that localized changes in ice thickness and/or ice extent occur, and may be larger than the global patterns.



## 4.1 MISMIP+

At the start of the MISMIP+ transient experiments, the total mass is 39 097 Gt, and the grounding line position is at x = 397
km, stable and within a bed trough (Fig. 1). Quantitative results are presented in Table 4 and Fig. 4, while the evolution of
individual ensemble members and of the probability density functions (PDFs) are displayed in Fig. 5. We base our analysis on
the leading-order statistical moments of the final ice mass PDF in each ensemble, focusing on the mean, standard deviation
($\sigma_E$), and skewness of the ensemble (Table 4, Fig. 4). For all ensembles, the distribution of the final ice mass is consistent
with a Normal distribution as assessed by the Shapiro-Wilk test at 5% significance level (Shapiro and Wilk, 1965) (Table 4).
In all the ensembles, the mean final ice mass is larger than the initial mass, but by less than $1\sigma_E$. In all ensembles, $\sigma_E$ is still
increasing at the end of the 500 years and the PDFs of final glacier state have not yet converged to statistical steady-states.
Thus, our analysis focuses on statistics of the distributions reached at a given moment in time.

| | Mean [Gt] | Mean relative change | Standard deviation [Gt] | Skew | Shapiro-Wilk p-value |
|---|---|---|---|---|---|
| $MISMIP_{\tau=1}$ | 39 121 | +0.06% | 41 | -0.173 | 0.31 |
| $MISMIP_{\tau=2.5}$ | 39 112 | +0.04% | 88 | 0.094 | 0.24 |
| $MISMIP_{\tau=10}$ | 39 122 | +0.06% | 366 | -0.080 | 0.72 |
| $MISMIP_{\tau=20}$ | 39 193 | +0.25% | 664 | -0.071 | 0.67 |
| $MISMIP_{\tau=50}$ | 39 184 | +0.22% | 1517 | -0.087 | 0.45 |

**Table 4.** Statistics of the final ice mass distributions for the 5 MISMIP+ transient ensembles. The relative change is calculated with respect
to the initial ice mass (39 097 Gt).

The cause of mean gains in ice mass for the MISMIP+ transient ensembles relates to the initial grounding line position in
a bed trough (Fig. 1), and to the form of the melt forcing imposed. Combined to the depth-dependent melt parameterization
(Eq. (2)), the initial geometric configuration causes an asymmetry in melt rate response: both advancing and retreating glaciers
move their grounding line depth to higher elevations, hence decreasing their respective mean melt rate at the grounding line.
This effect limits further retreat of retreating glaciers, while it favors further advance of advancing glaciers. Moreover, the bed
slope itself is asymmetric, being steeper upstream than downstream of the trough (Fig. 1). As a consequence, the mean final
glaciers are more advanced and with a larger ice mass than in the initial state. These results demonstrate that noisy forcing with
zero mean can drive non-zero mean response of the glacier, which is known as noise-induced drift (e.g., Hindmarsh and Le
Meur, 2001; Mikkelsen et al., 2018; Robel et al., 2018). Here, the noise-induced drift can be associated with asymmetries in
geometrical configurations and forcings, and with non-linearities in ice dynamics. In addition to the factors mentioned above,
there are other non-linearities at play that could contribute to the noise-induced drift: the floating area affected by melt varies
as the grounding line migrates, lateral stresses exerted from the grounded parts on the domain sides depend on the geometry,
and ice shelf buttressing depends non-linearly on ice shelf length, ice shelf thickness, and ice thickness at the grounding line
(Haseloff and Sergienko, 2018). The individual contributions of all these different factors on the noise-induced drift are difficult
to disentangle.



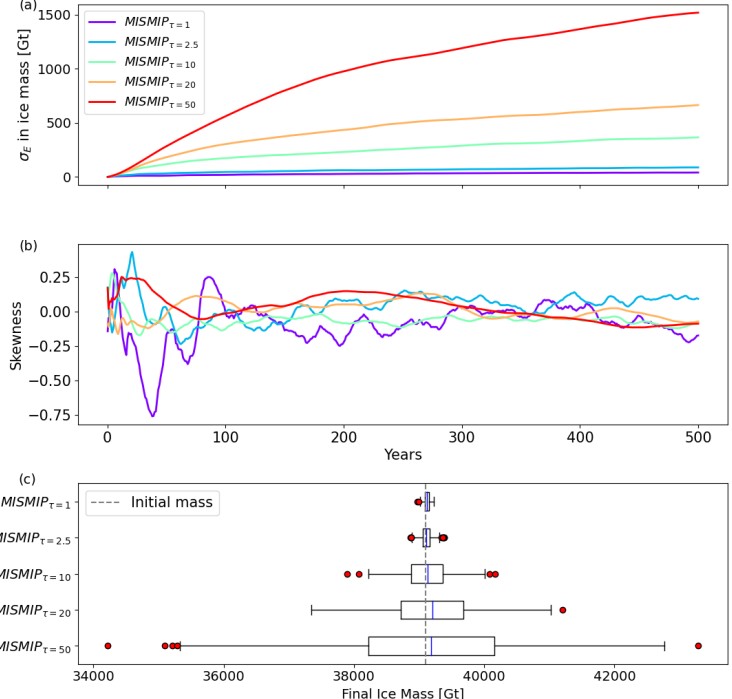

**Figure 4.** Evolution throughout the transient experiments of (a) the standard deviation, and (b) the skewness in total ice mass for the 5 MISMIP+ transient ensembles. (c) Boxplots of the final ice mass distributions. Red dots indicate ice masses beyond 1.5 times the inter-quartile range from the quartile.

Furthermore, the mean gain in ice mass is highest for the ensembles with higher $\tau$ values: $MISMIP_{\tau=20}$ and $MISMIP_{\tau=50}$. The spread in final ice mass, quantified by $\sigma_E$, increases sublinearly with $\tau$. For example, compared to $MISMIP_{\tau=1}$, $MISMIP_{\tau=10}$ yields a $\sigma_E$ 8.9 times larger while $MISMIP_{\tau=50}$ yields a $\sigma_E$ 36.9 times larger. The skewness of lower characteristic timescales

($\tau \leq 10$ yr) show more variability over time (Fig. 4); short time-persistence of the noise in melt rates frequently causes temporary glacier excursions that deviate strongly from the ensemble mean. In contrast, skewness of the higher characteristic timescale ensembles, $MISMIP_{\tau=20}$ and $MISMIP_{\tau=50}$, are smoother. With long time-persistence of the noise applied, unusual retreats or advances of the MISMIP+ glacier are slower to develop, but also to recover back towards the ensemble mean. While $MISMIP_{\tau=20}$ and $MISMIP_{\tau=50}$ show the largest spread in ice mass, their skewness is close to 0 throughout the 500 years,

indicating that large and small glacier outliers are approximately equally likely, and of similar magnitude in terms of mass difference with respect to the ensemble mean.

Roe and Baker (2016) derived an expression relating variability in glacier length, $\sigma_L(\tau)$, to $\tau$:

$$\sigma_L(\tau) = \sigma_L(\tau = \Delta t) \left( \frac{2\tau}{\Delta t} - 1 \right)^{1/2}, \tag{14}$$





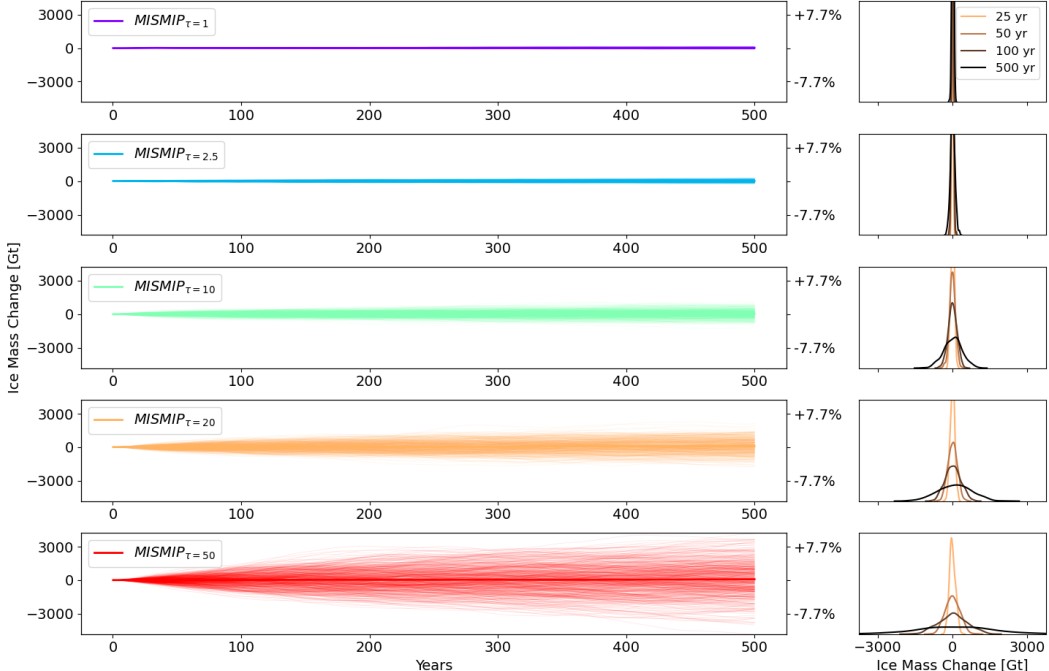

**Figure 5.** (left) Change in ice mass throughout the transient experiments for each ensemble member in the 5 MISMIP+ transient ensembles. Thick lines show the ensemble means. The right y-axis shows change relative to the initial ice mass. (right) Corresponding PDFs of the ice mass change after 25, 50, 100, and 500 years of simulations.

where we use $\Delta t = 1$ yr, and $\sigma_L(\tau = \Delta t)$ corresponds to glacier length variability under white noise forcing. This expression

was derived from an analytical three-stage mountain glacier model, and found to match results of a numerical flowline model (Roe and Baker, 2014, 2016). Eq. (14) predicts that the variability amplitude in glacier length increases with the square-root of the characteristic timescale of the noise. In our experiments, the relationship between $\tau$ and the standard deviation of the grounding line position is sublinear, but does not follow the square-root dependence predicted by Eq. (14). The absence of match with the theory predicted by Roe and Baker (2016) possibly illustrates that lateral shearing cannot be neglected in the

MISMIP+ configuration, that Eq. (14) does not hold on irregular bed slopes, and/or that Eq. (14) is only suited for glaciers where mass loss is controlled by SMB rather than by melt beneath buttressing floating ice. Ultimately, even idealized marine-terminating glaciers, such as the MISMIP+ configuration, include a much wider range of processes than the simple mountain glacier considered by Roe and Baker (2016).

### 4.2 Idealized Quarter Ice Sheet

The initial state of the IQIS is in equilibrium, thus any deviation from the initial state is attributable to the stochastic fluctuations imposed in our $IQIS_1$, $IQIS_2$, $IQIS_3$ and $IQIS_4$ transient experiments (Table 1). The initial ice mass is 350 715 Gt. Table 5 and



Fig. 6 show the quantitative results of each ensemble experiment, and Fig. 7 displays the evolution of each ensemble member and of the PDFs over time. According to the Shapiro-Wilk normality test (Shapiro and Wilk, 1965), all the final ice mass distributions are consistent with normality at the 5% significance level (Table 5). As in the MISMIP+ experiments, the $\sigma_E$

values of all ensembles are still increasing at the end of the 500-year transients, and the ice mass distributions have not reached a statistical steady-state yet (Fig. 6). For $IQIS_1$ (i.e., stochastic SMB) and $IQIS_4$ (i.e., stochastic $TF$), the mean final ice mass is -7.6% and +5.4% of their respective $\sigma_E$ away from the initial equilibrium ice mass. These small deviations of the mean state show that the stochastic fluctuations applied in these two ensembles do not cause significant noise-induced drift after 500 years.

|  | Mean [Gt] | Mean relative change | Standard deviation [Gt] | Skew | Shapiro-Wilk p-value |
|---|---|---|---|---|---|
| $IQIS_1$ | 350 668 | -0.01% | 620 | -0.088 | 0.40 |
| $IQIS_2$ | 352 750 | +0.58% | 567 | 0.064 | 0.78 |
| $IQIS_3$ | 352 779 | +0.59% | 1159 | 0.057 | 0.59 |
| $IQIS_4$ | 350 914 | +0.06% | 3660 | -0.21 | 0.12 |

**Table 5.** Statistics of the final ice mass distributions for the 4 IQIS ensembles. The relative change is calculated with respect to the initial ice mass (350 715 Gt).

In contrast, the results from $IQIS_2$ (i.e., stochastic $c_{rate}$) demonstrate strong evidence of noise-induced drift caused by stochastic white noise fluctuations in $c_{rate}$. The mean final ice mass is 2035 Gt larger than the initial mass, which corresponds to $> 3.5\sigma_E$, and to $> 0.5\%$ of the total ice mass. Since $c_{rate}$ cannot be negative at any given time step, $c_{rate}$ is set to 0 when $\epsilon_{c_{rate}}$ pushes it below 0. This causes a slight asymmetry in the distribution of the $\epsilon_{c_{rate}}$ applied. However, this effect is negligible as only 33 of the 500 ensemble members have $c_{rate} = 0$ at any time step, and all ensemble members have a final ice mass

larger than the initial mass. The latter is true even for ensemble members for which the stochastic realization of their $\epsilon_{c_{rate}}$ time series causes a total calving flux larger than the total calving flux of the deterministic scenario without stochastic perturbations. As such, it is the stochastic nature of our calving forcing and the ice sheet response to the stochastic forcing that cause the emergence of noise-induced drift, and not the total cumulative calving flux.

The perturbations $\epsilon_{c_{rate}}$ and $\epsilon_{TF}$, through its influence on $m_{trm}$ (Eq. 5), both impose stochastic noise on the frontal ablation.

One could therefore expect similar noise-induced drift to appear in $IQIS_4$, but there is no statistically significant evidence for this in our results after 500 years of transient simulation. Separate tests (not shown) demonstrate (i) that noise-induced drift appears in $IQIS_4$ if $\tau$ is reduced to 1 yr (i.e., annual white noise), and (ii) that noise-induced drift in $IQIS_4$ becomes statistically significant if more time is allowed for convergence of the PDF while keeping $\tau = 10$ yr. Thus, the longer persistence-time of the noise ($\tau = 10$ yr) requires longer timescales for the PDF to statistically converge, and thus for the noise-induced drift to be

significant with respect to the ensemble spread.

The results of $IQIS_3$ (i.e., stochastic SMB and $c_{rate}$) show a strong noise-induced drift, of approximately the same magnitude as $IQIS_2$ (Table 5). This is consistent with the same stochastic variability applied on $c_{rate}$ in both ensembles (Table 1). However, the additional stochasticity on SMB increases the spread, and the initial steady-state ice mass is within the final interquartile





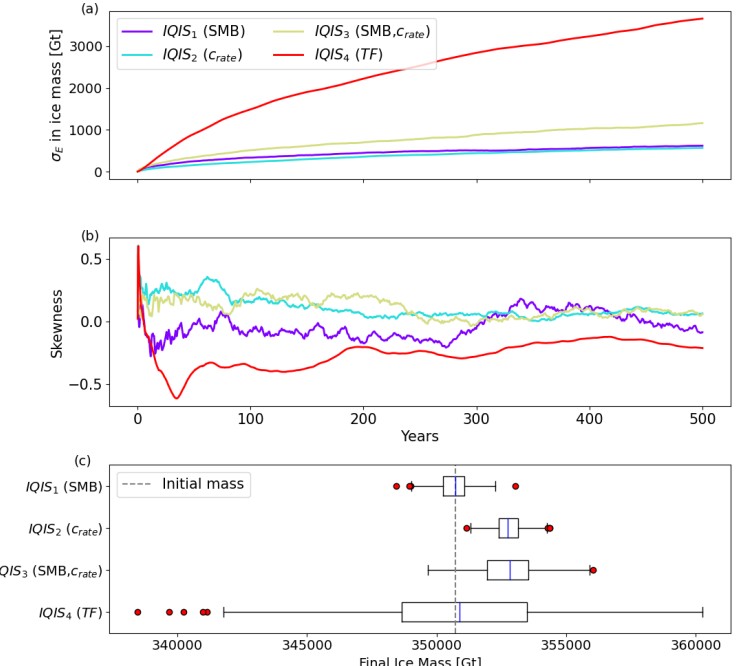

**Figure 6.** Evolution throughout the transient experiments of (a) the standard deviation, and (b) the skewness in total ice mass for the 4 IQIS ensembles. (c) Boxplots of the final ice mass distributions. The dashed line shows the initial ice mass, from the deterministic steady-state. Red dots indicate ice masses beyond 1.5 times the inter-quartile range from the quartile. In labels, variables between brackets denote variables with stochastic variability imposed (Table 1).

range, but still $> 1\sigma_E$ lower than the final mean. The $\sigma_E$ of $IQIS_3$ is very close to the sum of those of $IQIS_1$ and $IQIS_2$, and
is 98 Gt larger than if the latter were added in quadrature with the additional covariance factor accounting for the correlation between $\epsilon_{\mathrm{SMB}}$ and $\epsilon_{c_{rate}}$. As such, variability in the forcings does not have a one-to-one correspondence with variability in the final ice sheet mass.

Applying decadal variability in ocean thermal forcing ($IQIS_4$) leads to the highest spread in the final ice mass distribution. Furthermore, $IQIS_4$ also shows the strongest skewness throughout the 500-year period, and it is consistently negative (Fig. 6).
The negative skew is driven by a larger number of outliers in the lower tail compared to the higher tail (Fig. 6). As such, time-persistence in the stochastic ocean forcing causes more scenarios of extreme mass loss compared to the mean response, and the evolution of the skew contrasts with that of the MISMIP+ experiments. This confirms that response to stochastic forcing is not only asymmetric, but also that the asymmetry depends on the ice sheet state.

### 4.3   Greenland Ice Sheet

Results of the GrIS ensemble with correlated stochastic variability in SMB and *TF* forcings are shown in Fig. 8 and Table 6. We compare the variability in the ensemble to the drift in the deterministic control run. The initial ice mass is $2.743 \times 10^6$ Gt,



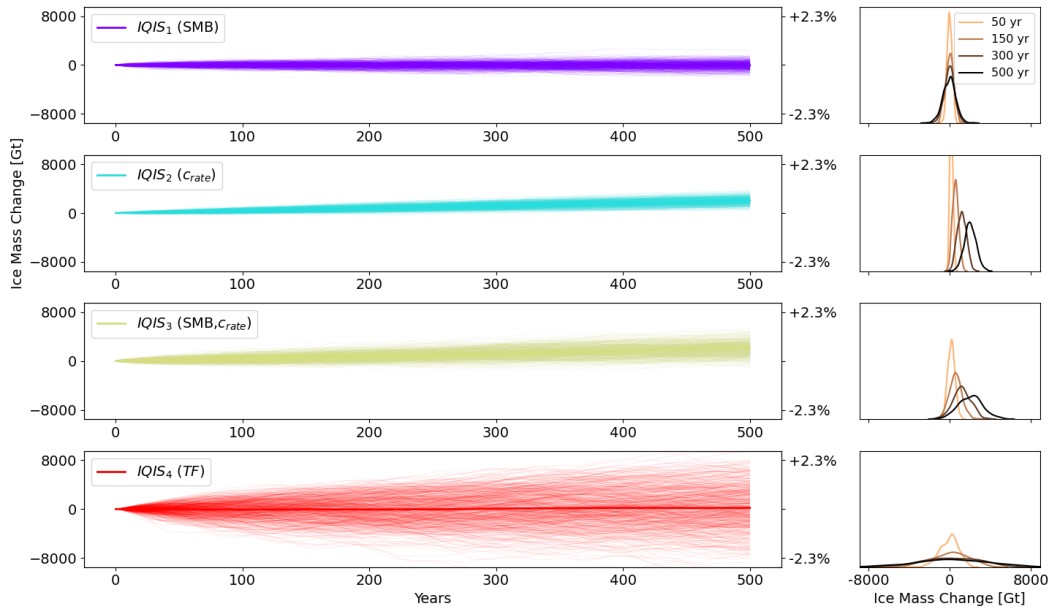

**Figure 7.** (left) Change in ice mass throughout the transient experiments for each ensemble member in the 4 IQIS ensembles. In labels, variables between brackets denote variables with stochastic variability imposed (Table 1). Thick lines show the ensemble means. The right y-axis shows change relative to the initial ice mass. (right) Corresponding PDFs of the ice mass change after 50, 150, 300, and 500 years of simulations.

and the deterministic drift over the 500 years is small but non-zero, up to +678 Gt (+0.02%) at the end of the 500-year run (Table 6). In contrast, the standard deviation in the final ice mass of the stochastic ensemble is 8907 Gt, thus >13 times larger than the deterministic drift. The mean final ice mass is 3175 Gt (-0.1 %) lower than the initial mass, which is thus $\sim \frac{1}{3}\sigma_E$ away

from the final mean, and remains within the final interquartile range (Fig. 8 and Table 6).

| Year | Mean [Gt] | Mean relative change | Standard deviation [Gt] | Skew | Shapiro-Wilk p-value | Deterministic drift [Gt] |
|------|-----------|---------------------|------------------------|------|---------------------|--------------------------|
| 125 | 2 742 227 | -0.04% | 5548 | -0.03 | 0.90 | +149 |
| 250 | 2 741 005 | -0.08% | 7099 | -0.22 | 0.20 | +315 |
| 375 | 2 740 441 | -0.10% | 7853 | 0.25 | 0.70 | +480 |
| 500 | 2 740 093 | -0.12% | 8907 | -0.14 | 0.91 | +678 |

**Table 6.** Statistics of the ice mass distributions of the GrIS ensemble after 125, 250, 375, and 500 years of simulation. The relative change and deterministic drift are calculated with respect to the initial ice mass (2 743 269 Gt).

The skew in ice mass varies between positive and negative phases, while the ensemble mean is strongly decreasing and the ensemble spread strongly increasing after 500 yr (Fig. 9 and Table 6). This shows that the distribution is still far from a statistical steady-state. We note that throughout the 500 years, the ensemble ice mass distribution remains consistent with





normality (Table 6). Among the 200 ensemble members, the highest and lowest final ice mass are +2.45 and -3.23 $\sigma_E$ away

from the mean, respectively, hence contributing to the final negative skew. This also means that the maximum range of final ice

mass in our ensemble amounts to 50 700 Gt, i.e, 1.8 % of the modeled GrIS. In the same way, after 500 years, the $\pm2\sigma$ range

of our ensemble is 1.3 % of the GrIS mass (Table 6). As illustrated by the distribution of the final ice mass, the response of

the GrIS to noisy forcings is asymmetric (Fig. 8), despite the imposed climatic fluctuations being symmetric around 0, and its

initial state being close to equilibrium. While the initial mass is within the $\pm1\sigma$ range of the mean final mass, only 37 % of the

ensemble members show a mass increase (Fig. 8). The persistence of an approximately linear decrease in the mean over the

500-year period suggests that the mean of the converged state would be even lower over the thousands of years likely necessary

to reach a statistical steady-state.

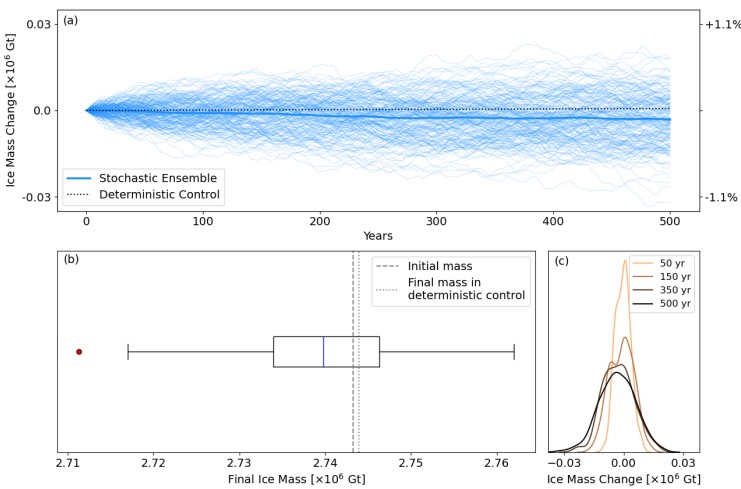

**Figure 8.** (a) Change in ice mass throughout the transient experiment for each ensemble member in the GrIS ensemble. The thick line shows the ensemble mean. The right y-axis shows change relative to the initial ice mass. (b) Boxplot of the final ice mass distribution. Red dots indicate ice masses beyond 1.5 times the inter-quartile range from the quartile. (c) PDF of the ice mass change after 50, 150, 300, and 500 years of simulations.

## 5 Discussion

Stochastic modeling is well-established in climate modeling (e.g., Porta Mana and Zanna, 2014; Berner et al., 2017). This

modeling approach is based on a rigorous mathematical framework, originating from stochastic differential equations and sta-

tistical physics (Majda et al., 2001; Franzke et al., 2015). In climate models, stochastic parameterization of internal variability

and unresolved processes has been shown to improve the skill of probabilistic forecasts, reduce systematic model errors, cap-

ture regime transitions, and modify the modeled response to external forcing (Berner et al., 2017; Palmer, 2019). StISSM v1.0

represents the first attempt to include stochastic parameterizations in large-scale ISMs. Our results for simplified and idealized



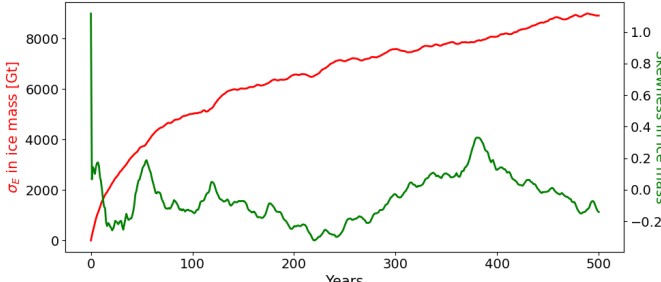

**Figure 9.** Evolution throughout the transient experiment of (red) the standard deviation, and (green) the skewness in total ice mass for the GrIS ensemble.

model experiments show that features similar to those observed in stochastic climate modeling occur in large-scale ISMs, such as noise-induced drift and a modified mean response to external forcing. These features are in agreement with previous simple model experiments (e.g., Mikkelsen et al., 2018; Robel et al., 2018). Our results also reveal that simple noise terms propagate to ice sheet evolution uncertainty in a complex way because of the high degree of non-linearity in ice sheet dynamics, and may be more nuanced than theory (Roe and Baker, 2016; Robel et al., 2019). As such, model simulations are required to quantify

the response of any particular glacier or ice sheet configuration to climate and internal variability, and this response cannot be trivially estimated a priori.

     Irreducible uncertainty is not quantified in current ice sheet model intercomparison projects (Goelzer et al., 2020a; Seroussi et al., 2020). Stochastic parameterizations facilitate quantification of the irreducible uncertainty component in ice sheet projections, which is an integral part of any model prediction. This component of uncertainty is expected to be larger in systems with

potential dynamical instabilities, in reality more pronounced than shown in our idealized experiments (Robel et al., 2019).

     StISSM v1.0 allows for stochasticity in variables which exhibit internal variability. The features of spatiotemporal correlation can be prescribed, as well as inter-variable correlations. Our model experiments show that the stochastic parameterizations implemented are functional, and can be used at ice-sheet scale. We have aimed at making StISSM v1.0 as user-friendly as possible, in such a way that any user familiar with ISSM should find the use of StISSM v1.0 straightforward. Ensemble runs

and parallelization allow for adequate sampling of irreducible uncertainty in model simulations. In general, a StISSM v1.0 simulation run with stochastic parameterizations uses additional computational resources that are negligible compared to a corresponding deterministic simulation. As in methods for estimating the role of parameter uncertainty in ice sheet evolution (e.g., Schlegel et al., 2018; Aschwanden al., 2019; Bulthuis et al., 2019), the main computational expense comes in running many simulations. The autoregressive modeling capabilities implemented offer a computationally fast way to generate climatic

forcings with prescribed statistical properties, thus sampling natural climate variability without the need to run a costly climate model for each ensemble member. In ISMs, variability in some glaciological processes such as calving and supra-, intra-, and sub-glacial water movement are particularly difficult to simulate; accurately resolving these processes remains elusive.



Stochastic parameterizations of such unresolved processes provide a way forward to better account for their impacts on large-scale and long-term ice dynamics.

A practical question that arises concerns the number of members needed per ensemble. Here, we have used 500 members for the MISMIP+ and IQIS ensembles, and 200 for the GrIS ensemble to limit computational expense. As the number of members increase, the statistics of the ensemble progressively converge to the statistics of the true underlying distribution. In other words, results from ensembles with increasingly more members converge to the results of an ensemble with infinitely many members. Convergence plots of the statistics of interest, such as the final mean, standard deviation, and skew in our case, shows

their progressive convergence, and is a useful tool in evaluating the number of members needed. We show such an analysis of our results in Appendix C, demonstrating adequate convergence of the ensemble statistics with 100 to 150 members in our experiments. It must be kept in mind that the number of members needed to represent the true distribution not only depends on non-linearities in the system that cause larger variability between members, but also on the time scale of stochastic variability imposed, and on the statistics of interest. For example, correctly estimating the 99th percentile of the distribution requires a

larger ensemble than for estimating the mean, as the effective number of members influencing this statistic is smaller.

    With this first version of a large-scale stochastic ISM, several future research priorities can be identified. First, the statistics of variability in the unresolved processes need to be evaluated in order for the stochastic parameterizations to capture their impacts on ice dynamics accurately. This could be achieved via theoretical, observational, and/or high-fidelity modeling studies (e.g., Bassis, 2011; Hewitt, 2013; Emetc et al., 2018; Christensen, 2020; Åström al., 2021; Hu and Castruccio, 2021).

We anticipate providing workflows for estimating these parameters in future studies. Such workflows will provide calibrated covariance matrices and autoregressive parameters ready to use for StISSM v1.0 simulations. Better representing the statistics of stochastic forcings may necessitate future improvements in StISSM to allow for non-Gaussian variability or other statistical time series models. Second, there is a computational limitation: sampling irreducible uncertainty requires ensemble runs with a statistically representative number of ensemble members. As ice sheet-wide ISM simulations remain computationally ex-

pensive, ensemble runs thus require a compromise between number of members and model resolution or number of simulated processes.Third, while we have implemented stochastic capabilities for subglacial water pressure and verified their functionality in test experiments, this study does not discuss simulations with stochasticity imposed on this variable. Realistic benchmark model experiments first require work on choosing an appropriate sliding law and constraining water pressure variability in time. Similarly, while the autoregressive models for climate forcing are flexible approximations of climate variability, they

need to be calibrated against climate records or model outputs (Chylek et al., 2012; Castruccio et al., 2014). Finally, a major challenge is the validation of a stochastic parameterization. Current ISMs calibrate model parameters to match observational datasets that are much shorter than the response timescale of ice sheets. As such they may compensate for not representing the effects of variability in some forcings by using biased parameter values. For example, the noise-induced drift effect in our IQIS results using variability in calving rates could be compensated for by a biased estimate of the commonly-used ice tensile

strength parameter, which is generally tuned to values much lower than minimal values from measurements (Petrovic , 2003; Amaral et al., 2020; Ultee et al., 2020; Choi et al., 2021; Hillebrand et al., 2022).





Our results show that ice sheets need a long period of time to converge to a statistical steady-state in the presence of noisy forcing. This raises the question of how zero-mean variability during spin-up could affect the ensuing modeled initial state of an ice sheet. We have used realistic inter-annual and decadal variability in SMB and ocean forcing for our GrIS experiment, reasonably representative of the noisy climate under which the Greenland ice sheet evolves. After 500 years of our synthetic experiment, this resulted in a $\pm 2\sigma$ spread equivalent to 1.3 % of the initial mass, and a decrease in the mean mass of -0.1 %. Spin-up procedures assuming that climate forcing is constant in time, or without high-frequency noise, lead to subsequent changes in transient experiments that are partially caused by the ice sheet adapting to the abrupt presence of higher-frequency variability.

## 6 Conclusions

This study has described the development, implementation, and testing of StISSM v1.0, the first stochastic large-scale ice sheet model. Variables with implemented stochastic parameterizations in this first version encompass climate forcing and glaciological processes that are unresolved at the spatiotemporal resolution of ice sheet models: SMB, ocean forcing, calving and subglacial water pressure. Stochastic climate forcing captures the irreducible uncertainty in climate predictions, and how it translates into projected ice sheet mass balance uncertainty. Using stochastic parameterizations for unresolved glaciological processes facilitates the quantification of the impacts of internal variability in such processes on ice dynamics. StISSM v1.0 also includes built-in statistical models for generation of stochastic variability in SMB and oceanic forcing, represented as autoregressive processes. The statistics of the stochastic variability and of the autoregressive climate models are prescribed by users, and can thus be adjusted to particular user needs.

We have tested the stochastic capabilities in idealized, synthetic model experiments. These tests have demonstrated that the stochastic parameterizations are functional, and can be up-scaled to realistic ice sheet configuration. Our results show that stochastic forcings cause responses of the ice sheet system in line with those observed in stochastic climate and ocean model experiments. For example, stochastic forcing not only causes variability in the final state, but also asymmetry in the response, noise-induced drift, and long timescales needed for ice sheet state convergence. Even in the simple experiments proposed here, the features of the response are complex and cannot be quantified without running ensemble simulations. The response of a particular system is sensitive to the type of forcing, to the geometric configuration, and to the intrinsic non-linearity of ice dynamics. Our results thus raise important questions about representing fluctuating processes with constant deterministic parameterizations, about neglecting high-frequency climatic noise, and about ice sheet model initialization performed without imposing variability.

Our strategy for the development of StISSM v1.0 allows for potential future extensions of stochastic capabilities to other variables in a straightforward manner. In the future, calibration work will be needed to constrain the statistical models for climate forcing, as well as the variability in unresolved glaciological processes such as calving and hydrology. Such an effort will require combining observations, theory, and results from high-fidelity model experiments to understand the internal spatiotemporal variability of processes of interest. Our implementation allows for any spatial, temporal, and inter-variable correlation



features. StISSM v1.0 thus provides a robust modeling framework to quantify the impacts of forcings with internal variability
on ice sheet mass balance.

*Code and data availability.*   The stochastic schemes evaluated here are currently implemented in the public release of ISSM. The code can be
downloaded, compiled, and executed following the instructions available on the ISSM website: https://issm.jpl.nasa.gov/download (last ac-
cess: 21 May 2022). The public SVN repository for the ISSM code can also be found directly at https://issm.ess.uci.edu/svn/issm/issm/trunk
and downloaded using username "anon" and password "anon". The version of the code for this study, corresponding to ISSM release 4.19,
is SVN version tag number 27017. The documentation of the code version used here is available at https://issm.jpl.nasa.gov/documentation/
(last access: 21 May 2022). The simulation results, and the scripts to reproduce all the figures are available as a Zenodo dataset:
https://doi.org/10.5281/zenodo.6720259

## Appendix A:  GrIS spin-up state

In this section, we briefly provide additional details concerning the GrIS configuration at the end of the spin-up (Sect. 3.3).
The fields of ice thickness and ice velocities are displayed in Fig. A1c and A1d. These can be compared to the initial GrIS
configuration matched to observational datasets (Sect. 3.3) in Fig. A1a and A1b. The differences in ice thickness and velocities
between both configurations are shown in Fig. A1e and A1f. We note three main differences (i) a major thinning at the margins
of the North-East Greenland Ice Stream (NEGIS), (ii) thickening in the South-West, and (iii) velocity changes at the simulated
outlet glaciers typically ranging between -200 and +200 m/yr. The thickness differences across the GrIS are mostly caused by
the geometrical adjustment of the ice sheet away from its observed state, which in turn drives SMB feedback processes via
the lapse rates applied (Table 2). In contrast, geometrical changes due to frontal ablation and migration allowed in the second
phase of the spin-up (see Sect. 3.3) are small, because the calving rates have been calibrated for this purpose (Table 3). At the
margins of the NEGIS, thinning is caused by an imbalance between ice flow and initial thickness, which in turn causes the
SMB to decrease. There is a noticeable increase in ice velocities just upstream of the margin of NEGIS (Fig. A1f), but with a
minor impact on ice flow due to the strong thinning in that area (Fig. A1e).

In Fig. A2, we show the changes in ice thickness and velocities over the last 200 years of spin-up. Note that the second phase
of the spin-up, in which we resolve the 11 outlet glaciers, has a total duration of 1000 years; Fig. A2 displays the changes
between year 800 and year 1000 of that phase. The changes are relatively small, with maximum magnitudes in thickness
change and velocity change of $\sim$40 m (rate of 0.2 m/yr) and $\sim$35 m/yr (rate of 1.8 m/yr$^2$), respectively. Only glacier XI
(Heilprin) still shows small patterns of frontal migration and thinning after 800 year of simulation with frontal migration
activated (Fig. A2). The drift due to imperfect equilibrium of the initial state is quantified with a control run (Sect. 3.3). We
consider this drift sufficiently small (Sect. 4.3), and thus with negligible impact on the transient results driven by stochasticity.
A larger drift would obscure the interpretation of results because it would require the assumption that changes due to drift and
stochastic variability in forcings can be separated linearly.





**Figure A1.** Initial configuration from observational datasets: (a) ice thickness, and (b) ice velocities. Configuration of the steady-state at the end of the spin-up: (c) ice thickness, and (d) ice velocities. Total differences between the steady-state and the initial configurations: (e) ice thickness ((c)-(a)), and (f) ice velocities ((d)-(b))



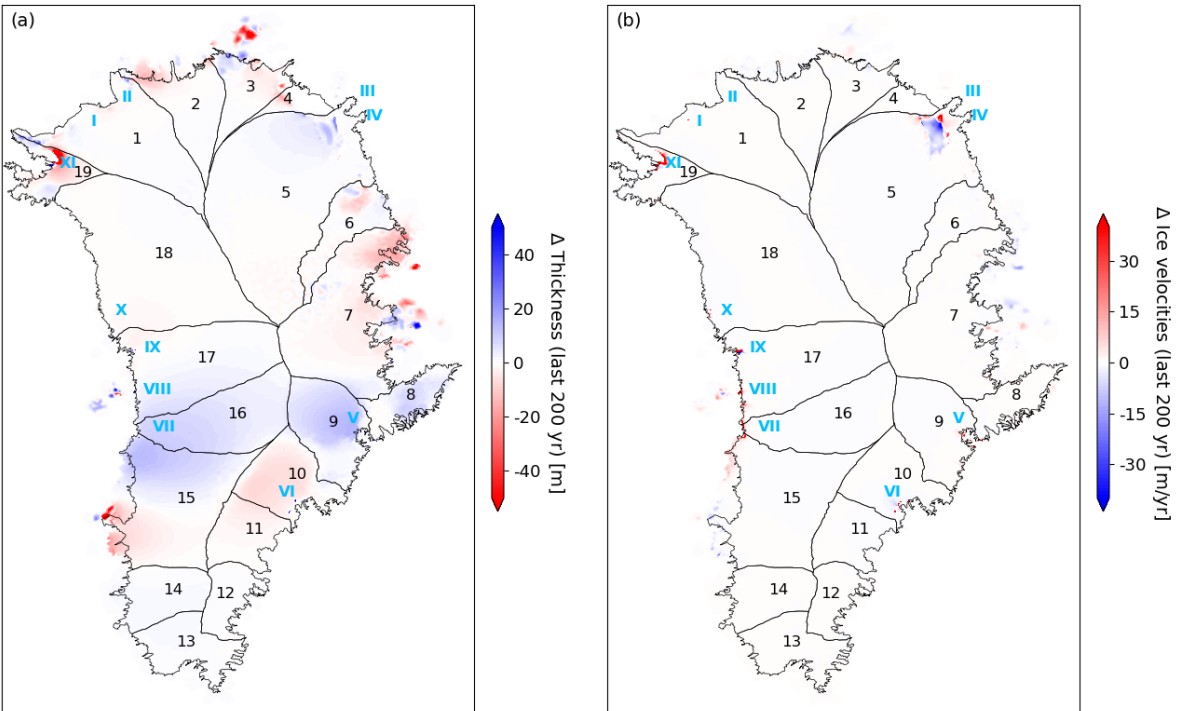

**Figure A2.** Differences in (a) ice thickness and (b) ice velocities over the last 200 years of the spin-up to a GrIS steady-state. Arabic numerals (black) show the individual basins. Roman numerals (cyan) show the outlet glaciers where ice front movement is simulated and ocean melt parameterized.

## Appendix B:  Correlation matrix for the GrIS transient ensemble

The correlation matrix relating noise terms for both SMB and *TF* in all basins of the GrIS stochastic transient runs is specified by Eq. (B1). In Eq. (B1), subscripts $i$ and $j$ denote basins that constitute individual spatial dimensions in the covariance matrix. From the correlation matrix given by Eq. (B1), the covariance matrix is subsequently obtained by left- and right-multiplying it with the diagonal matrix of the standard deviations (see Eq. (13)).


$$
\begin{cases}
r\left(\epsilon_{\text{SMB},i}, \epsilon_{\text{SMB},j}\right) = 0.6 & \text{if } i \text{ is neighbor of } j \\
r\left(\epsilon_{TF,i}, \epsilon_{TF,j}\right) = 0.6 & \text{if } i \text{ is neighbor of } j \\
r\left(\epsilon_{\text{SMB},i}, \epsilon_{TF,j}\right) = -0.6 & \text{if } i = j \\
r\left(\epsilon_{\text{SMB},i}, \epsilon_{\text{SMB},j}\right) = 0.5 & \text{if } i \text{ is not a neighbor of } j \\
r\left(\epsilon_{TF,i}, \epsilon_{TF,j}\right) = 0.5 & \text{if } i \text{ is not a neighbor of } j \\
r\left(\epsilon_{\text{SMB},i}, \epsilon_{TF,j}\right) = -0.4 & \text{if } i \neq j
\end{cases}
. \tag{B1}
$$





## Appendix C: Convergence of the statistics

In this section, we analyze how the statistics of interest converge as the number of members per ensemble increases. The statistics of interest are the mean, the standard deviation, and the skew in final ice mass. The analyses of convergence for the

MISMIP+, IQIS, and GrIS ensembles are shown in Figs (C1), (C2), (C3), respectively. To generate these figures, an initial random sample of five ensemble members is selected, and the statistics for this sample are computed. Iteratively, we add a random sample of five members to this initial sample, and compute the statistics at each iteration. The process is performed until all the ensemble members are included in the analysis. A qualitative evaluation of the figures show that statistics exhibit variability generally until 100 to 150 members are included. Beyond 150 members, the statistics show adequate convergence.

We show $\sigma_E$ relative to $\sigma_E$ of the full corresponding ensemble, because variability in absolute $\sigma_E$ values depends on the full ensemble $\sigma_E$. As mentioned in Sect. 5, we emphasize that the number of members needed for convergence will vary depending on the ice sheet system investigated, on the time scale of stochastic variability, and on the statistics of interest.

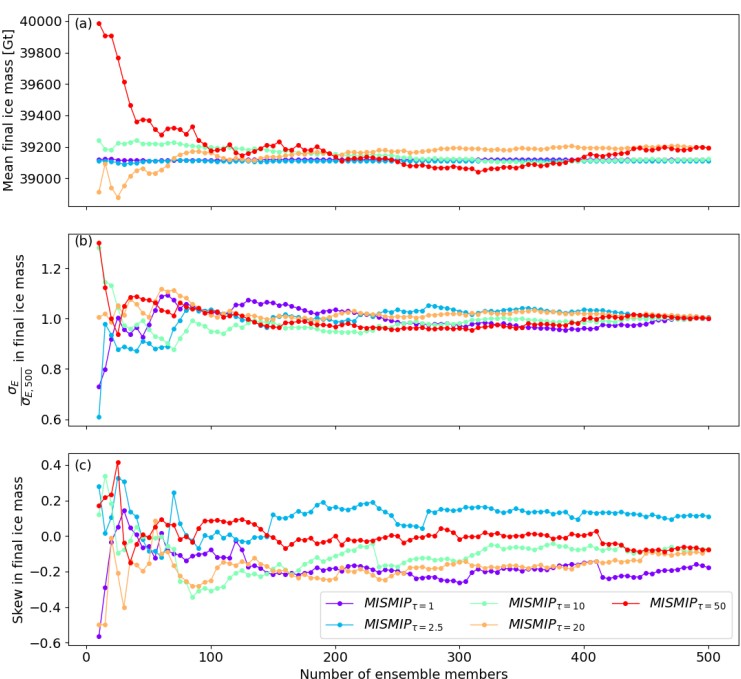

**Figure C1.** Convergence of the (a) mean, (b) standard deviation, and (c) skew in final ice mass for the MISMIP+ transient ensembles as a function of number of ensemble members. For (b), we show the standard deviation relative to the standard deviation of the full ensemble with 500 members.





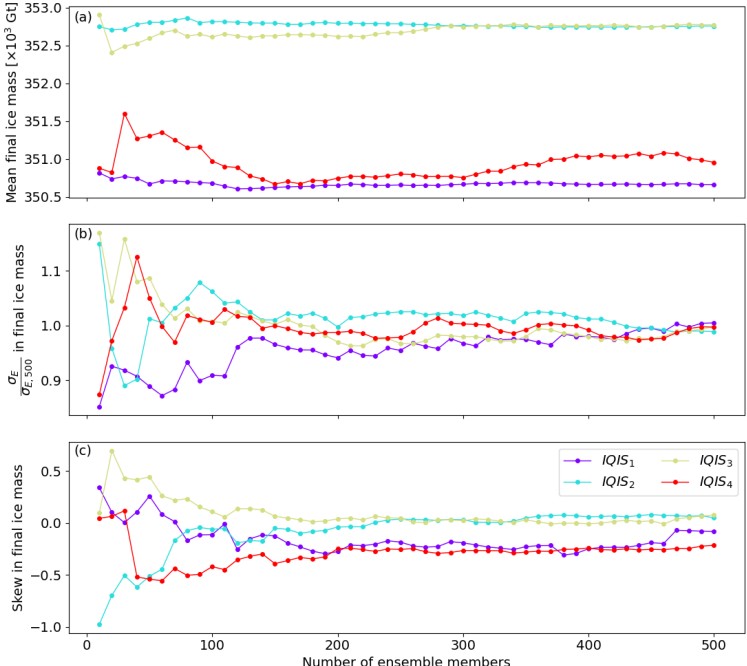

**Figure C2.** Convergence of the (a) mean, (b) standard deviation, and (c) skew in final ice mass for the IQIS transient ensembles as a function of number of ensemble members. For (b), we show the standard deviation relative to the standard deviation of the full ensemble with 500 members.

*Author contributions.* VV led the model development, performed the model experiments, and led the writing of the manuscript. AR supervised the work. HL contributed to the model development. VV, AR, HL, and AT conceived the study. LU and VV have worked on the

formulation and calibration of the statistical models. All authors provided comments and suggested edits to the manuscript. All authors, and this study, are part of the Stochastic Ice Sheet Project, aimed at understanding ice-sheet sensitivity to variability.

*Competing interests.* The authors declare that they have no conflict of interest

*Acknowledgements.* This work was funded by a grant from the Heising-Simons Foundation. HS was also funded by the NSF Navigating the New Arctic Program. Computing resources were provided by the Partnership for an Advanced Computing Environment (PACE) at the

Georgia Institute of Technology, Atlanta. We acknowledge HPC assistance from Fang (Cherry) Liu. We thank Mathieu Morlighem and Justin Quinn for providing helpful advice about ISSM. We thank Kevin Bulthuis for the implementation of the random number generator. We thank all developers of ISSM for their continuing work on model development. VV thanks John Christian for his interest in the study, and for insightful discussions about ice sheet sensitivity to variability.




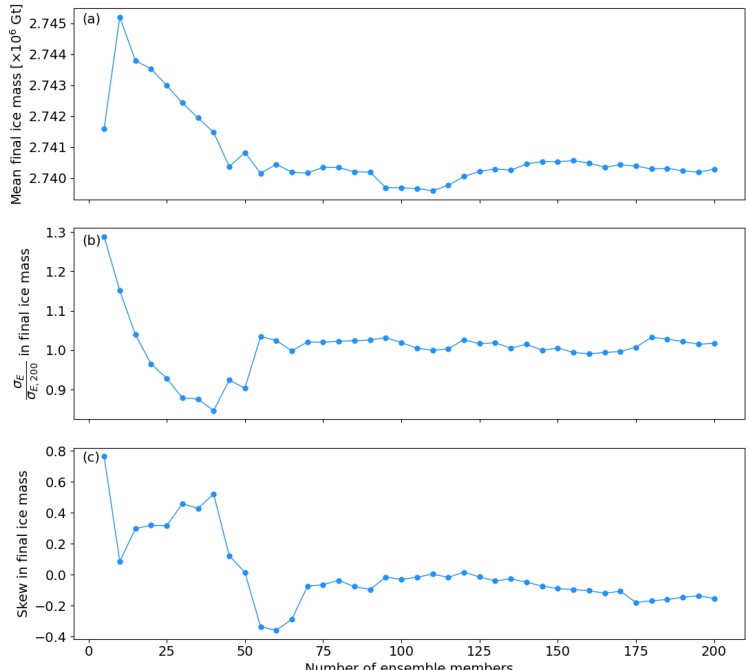

**Figure C3.** Convergence of the (a) mean, (b) standard deviation, and (c) skew in final ice mass for the GrIS transient ensemble as a function of number of ensemble members. For (b), we show the standard deviation relative to the standard deviation of the full ensemble with 200 members.

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
