# Peer review of "The Stochastic Ice-Sheet and Sea-Level System Model v1.0 (StISSM v1.0)"

_EGUsphere, 2022_

## Author Comment (AC3)

We thank Reviewer 1 for their interest in our work, and for providing constructive comments and suggestions on the manuscript. We have worked on the comments, and revised the manuscript. We have addressed all the "General comments" of Reviewer 1, and we have followed their suggestion for almost all their "Editorial comments". This document consists of a point-by-point reply to all the comments. In this document, the comments from Reviewer 1 are in blue, while our responses are in black. New text that has been included in the manuscript is in *black italic*.

A revised version of the manuscript will accompany this response. We will also upload a "tracked changes" version of the manuscript where all the modifications are highlighted. Line numbers in this document refer to line numbers of the revised manuscript without tracked changes.

This paper describes a new feature developed for the Ice-sheet and Sea-level System Model (ISSM), which adds stochastic parameterizations for particular forcings and model parameters. The new model framework is called The Stochastic Ice-Sheet and Sea-Level System Model v1.0 (StISSM v1.0). Overall, the paper is very clear in its structure and explanation of the purpose of the new functionality, an overview of how to use the new functionality, as well as interpretation of some initial experiments that used the new functionality. The authors should be commended for explaining a complicated concept (adding stochasticity to an already existing model framework) in a very clear and straightforward way. Additionally, the paper makes use of existing "standard" ice sheet model setups (e.g., MISMIP+ and IQIS) and makes good connections with prior related literature (e.g., the comparison with Roe and Baker, 2016).

I have some general comments as well as editorial comments further below, amounting to minor revisions needed before resubmission. I will emphasize again that the paper is very well written and presented in a clear manner. The StISSM model will be very impactful for future studies and I am pleased to see this significant step forward in ice sheet modeling.

We thank Reviewer 1 for their interest in our work.

General comment 1:
In the introduction, I suggest some additional text to state that, although StISSM will alleviate the need to run large ensembles of climate models if it is possible to correctly parameterize the structure of internal climate variability from other sources of information, such as observations. In this current manuscript draft, the wording in the introduction too strongly claims that StISSM will eliminate the need to run large GCM ensembles. I think there still may be a need, although that need could very well be reduced by StISSM. This will also connect nicely with the discussion text on lines 553-554.

We agree with Reviewer 1. However, significant advances have been made to statistically emulate climate model output based on a relatively small number of GCM runs (e.g., Castruccio and Stein, 2013; Castruccio et al., 2014; Bao et al., 2016). This raises the prospect to use statistical models within ISMs to reproduce the characteristics of climate variability and capture irreducible climatic uncertainty. We have included in the Introduction the reservation about the need to still run GCM ensembles for calibrating such models. We have also added a sentence about the prospect to use simpler statistical models to reproduce output representative of GCM outputs (l 93-98):

*Statistics that determine the magnitude and the spatiotemporal dimensions of variability should be provided by users, and thus constrained from theory, observations or other model simulations (Bassis, 2011; Chylek et al., 2012; Castruccio et al., 2014; Christensen, 2020; Hu and Castruccio, 2021). The latter option shows that StISSM v1.0 does not completely eliminate the need to run large climate model ensembles to constrain climatic variability. However, statistical models can be calibrated to a small number of climate model runs, and implemented in StISSM v1.0, in order to reproduce the characteristics of climate model output (Castruccio and Stein, 2013; Castruccio et al., 2014; Bao et al., 2016).*

General comment 2:
In the future, is it planned to add functionality to specify different stochastic time steps for different input variables (if they are uncorrelated)? I suggest adding some text to either the methods or the discussion sections to touch on this.

This is a future development that can be considered. As pointed out in the question, it is not straightforward to implement for correlated variables, because noise terms cannot be generated independently in this case. We have included this prospect in the manuscript (l 169-171):

*At this stage, StISSM v1.0 uses an identical stochastic time step for all variables modeled with additive Gaussian white noise (Eq. (1)), but implementing different stochastic time steps is a possible avenue for future development.*

General comment 3:
I suggest making the connection between "y" and "eta" more clear. I think it's good to have two separate symbols to make it clear that one is coming from an autoregressive process. But it would be good to state very explicitly: "At each simulation time step, a value for either y or \eta is calculated and used for that particular time step and subsequent simulation time steps until the next stochastic update." That wording is a little clunky and can definitely be improved. My suggestion is just to make it clear that "y" and "eta" represent a similar thing: the realization of a random variable that is used as the value for a particular forcing or parameter by ISSM.

We agree with the Reviewer that the difference between y and eta might have been confusing for readers unfamiliar to stochastic process. We now specify the difference between both explicitly before introducing the equation for autoregressive processes (l 213-215):
*We use the notation $\eta$ to represent a variable governed by a stochastic AR process, in contrast to y, which is governed by an additive Gaussian white noise process (see Eq. (1)).*

In Sect. 2.3, we have also clarified that both y and eta are ISSM variables that are represented as random processes in StISSM v1.0 (l 239-240):
*It should be noted that $\eta$ (Eq. (7)) and y (Eq. (1)) are both the realization of a random process used as an ISSM variable, with the former being an autoregressive stochastic process and the latter an additive Gaussian white noise process.*

General comment 4:
I suggest changing the presentation of the changes in ice mass in Section 4 from showing the initial mass and final mean masses in each ensemble to showing the mean changes in mass. In other words, show just the differences between the initial mass and the final mean masses in Gt, as well as the percent changes (as you have already shown). I don't see the need to show the initial and final masses themselves; the differences will illustrate the results more clearly. This would also make it easier to compare the mean mass change against the deterministic drift in Table 6. Additionally, please change how this is shown in the figures (e.g., Fig 4c, Fig 6c, Fig 8b).

We have updated all the figures, including the box plots, to show the ice mass change. We have also included an additional column in all the Tables of Sect. 4 that shows the total mass change [Gt] in addition to the column showing the relative mass change [%].

General comment 5:
The "Code and data availability" section states that "the simulation results, and the scripts to reproduce all the figures are available" on Zenodo. Are the scripts user to initialize, configure, and run the ISSM simulations also available there. The policy states that "preprocessing, run control and postprocessing scripts" and, I do see

postprocessing scripts in the Zenodo archive but I don't see preprocessing and run control scripts. If these are there, please ignore this comment. If they are missing, please provide these in the same Zenodo repository.

The reviewer is correct. The previous Zenodo dataset did not include input files, preprocessing scripts, and run control scripts. We have created a new Zenodo dataset that includes all these files and scripts, in addition to the ensemble results of our study and the python scripts used for the statistical analyses and for making the figures. The run scripts for the deterministic spin-up to steady-state of each experiment are available. In addition, we provide the saved final steady-state files such that the transient ensemble runs can be executed by readers without running the full spin-up again. We have updated the Code and Data availability section (please note that these changes are not highlighted in the "tracked changes" version of the manuscript for some unknown reason) (l 653-655):

*The simulation results, the scripts to reproduce all the figures, the scripts to perform statistical analyses, as well as all the input files, preprocessing, run control, and postprocessing scripts to reproduce the simulations are available as a Zenodo dataset:*
*https://doi.org/10.5281/zenodo.7144993*

Editorial comments
line 9: Change "of" to "for"
Done
line 29: It'll be a mouthful but I would spell out "CMIP6" here.
Done
line 32: Change "inclusion" to "selection"
Done
lines 82-84: This statement is brought up as motivation for the paper but it's not really addressed: "Finally, climate model simulations are generally not coupled to ISMs, which neglects possible impacts from ice-sheet changes on the climate system, such as surface elevation changes and modified ice discharge into the ocean." I suggest removing this from the intro because it doesn't directly motivate the need for a stochastic ice sheet model. Alternatively, if you'd like to keep it in, I suggest adding discussion text on how StISSM can help address the coupling issue.
We agree with the reviewer and have removed these statements. Please note that the autoregressive SMB scheme allows for a feedback process by using lapse rates. However, this is not necessarily related to the stochastic schemes, and could be implemented in a fully deterministic ISM. For this reason, we have preferred to follow the advice from Reviewer 1 and have removed the statements.
line 96: Possibly change "underline" to "emphasize" for clarity
Done
line 101: Add "The new ..." before the start of this first sentence to make it clear that this is referring to the new stochastic functionality within ISSM and not something that had already existed.
Done
line 106: I might suggest changing "ocean forcing" to "frontal ablation" or something like that. I think of "ocean forcing" as a climatic forcing (i.e., ocean thermal forcing), whereas the way that you convert ocean forcing to frontal melt (via parameterization) is a glaciological process.
Here, we have preferred not to follow the recommendation of Reviewer 1. There are different stochastic schemes related to ocean forcing. First, TF can be represented as an autoregressive process. TF is the temperature above freezing point and thus really represents a climatic forcing in ISSM simulations. Second, ice shelf melt rates can also be represented as an autoregressive process (or as an additive Gaussian white noise process). Ice shelf melt rate is indeed a parameterization of a glaciological process rather than a climatic forcing. Because stochastic variables related to oceanic conditions encompass both "raw" climatic forcing and parameterization of ocean melt, we prefer to use the broad term "ocean forcing" to give an overview of the processes targeted by

stochasticity. We believe that our descriptions in the Methods section make clear which precise variables are modeled as stochastic processes.

lines 347-348: The extrapolation of C_B is a bit unclear to me. Is this needed because the ice sheet will grow in extent during the transient simulation to get to steady-state? In other words, this is an extrapolation of C_B beyond the extent of the present-day ice sheet, where there are no ice velocities available to invert for C_B, correct? Please clarify.

The reviewer is correct that the use of a regression is necessary to infer realistic C_B values beyond the present-day ice sheet extent. In addition, at the very margins of the ice sheet, ice is thin and observed velocity gradients are large. This can cause spurious unrealistic patterns in C_B when doing an inversion. Thus, our regression is used to extrapolate C_B in any area with ice thickness less than 500 m to avoid this problem. We have clarified this in the manuscript (l 372-375):

*We use a linear regression of C_B on bed elevation to extrapolate the field in areas covered by less than 500 m of ice. This avoids spurious patterns in C_B in marginal areas where observed velocity gradients are large, and also allows extrapolation of C_B in ice-free areas where the ice sheet can extend during model simulations.*

line 360: "Free-flow boundary condition" is the same as "Neumann boundary condition", correct? If so, please state this.

Free-flux boundary conditions are not equivalent to Neumann boundary conditions. Instead, it is a boundary condition that allows the ice flux at the boundaries to be adjusted depending on the upstream flux in order to preserve the margin at the same position. Please also note that this approach was used in the ISMIP6 intercomparison experiment (Goelzer et al., 2020). We have clarified our description of the boundary conditions in the manuscript (l 387-388):

*In the first phase, we fix the ice sheet margin positions and implement free-flux boundary conditions at the ice margins, meaning that boundary ice fluxes adjust to incoming fluxes to keep margins fixed in space.*

line 383: I suggest changing "to quantify the minimal amount of deterministic model drift" to "to quantify the amount of deterministic model drift, which is minimal", if that is indeed what is meant here.

Done

Figure 3: Please add ticks and axis labels and also make sure that the axes are equal so that Greenland doesn't appear stretched in one direction or the other.

Done

line 408: Please add a very brief explanation for what the Shapiro-Wilk test signifies and how to interpret the p-value.

Done (l 436-439):

*We use the Shapiro-Wilk test to evaluate if the final ice mass PDFs are consistent with a Normal distribution (Shapiro and Wilk, 1965). This test measures the fit between standard normal quantiles and the ordered and standardized ice mass values of the ensemble. It has also been shown more powerful than many commonly-used normality tests (Razali and Wah, 2011).*

line 410: Please provide support for the statement: "and the PDFs of final glacier state have not yet converged to statistical steady-states." Is this determined by looking at the changes in the PDF statistics (mean, std dev, skew) over the last X years? Please specify.

We now explicitly state that the trends over the last 50 years of the 500-year simulations in both the ensemble mean and the ensemble standard deviation are significant (all p-values are below 0.001 in our experiments). We believe that the existence of a significant trend in the first two moments (mean and variance) of the ensemble results demonstrate that the PDFs have not reached statistical convergence. See l 441-443, l 486-488, and l 529-531.

line 413: Change "combined to" to "due to"

Done

line 570: Would it be fair to make "experiments" more specific by changing to "laboratory experiments"? If so, please make that change.

We now specify (l 608-609):

*(…) values much lower than minimal values from field- and laboratory-derived measurements (…)*

line 593: I suggest elaborating on the statement that stochastic forcing causes "asymmetry in the response." From the experiments presented, it seems to me that there is asymmetry during the transient but that the asymmetry decreases towards the end of each simulation and, as demonstrated by the Shapiro-Wilk p-values, ends up being fairly close to a symmetric normal distribution. If I am misinterpreting, it is because of my lack of familiarity with the Shapiro-Wilk test and that should be addressed in the paper (I have a comment about this above). But if what I wrote is correct, I suggest expanding this conclusion to state something similar to what I have suggested.

We agree that the term "asymmetry" is confusing in the way it was previously used. We do not necessarily refer to the final ice mass distribution being asymmetric, but rather that a zero-mean forcing can cause a non-zero-mean response as well as time-varying tendencies in the response of an ice sheet. To avoid confusion related to the term "asymmetry", which readers could associate to the PDFs themselves, we have rephrased this part of the Conclusion (l 632-633):

*(…) stochastic forcing not only causes variability in the final state, but also non-zero tendencies in the response, noise-induced drift, and long timescales needed for ice sheet state convergence.*

Thank you for your constructive comments.
Vincent Verjans, on behalf of all authors

References in this response

Bao, J., McInerney, D. J., and Stein, M. L.: A spatial-dependent model for climate emulation, Environmetrics, 27, 396–408, https://doi.org/10.1002/env.2412, 2016.

Castruccio, S. and Stein, M. L.: Global space-time models for climate ensembles, Ann. Appl. Stat., 7, 1593–1611, https://doi.org/10.1214/13-AOAS656, 2013.

Castruccio, S., McInerney, D. J., Stein, M. L., Crouch, F. L., Jacob, R. L., and Moyer, E. J.: Statistical emulation of climate model projections based on precomputed GCM runs, J. Climate, 27, 1829–1844, https://doi.org/10.1175/JCLI-D-13-00099.1, 2014.

Goelzer, H., Nowicki, S., Payne, A., Larour, E., Seroussi, H., Lipscomb, W. H., Gregory, J., Abe-Ouchi, A., Shepherd, A., Simon, E., Agosta, C., Alexander, P., Aschwanden, A., Barthel, A., Calov, R., Chambers, C., Choi, Y., Cuzzone, J., Dumas, C., Edwards, T., Felikson, D., Fettweis, X., Golledge, N. R., Greve, R., Humbert, A., Huybrechts, P., Le clec'h, S., Lee, V., Leguy, G., Little, C., Lowry, D. P., Morlighem, M., Nias, I., Quiquet, A., Rückamp, M., Schlegel, N.-J., Slater, D. A., Smith, R. S., Straneo, F., Tarasov, L., van de Wal, R., and van den Broeke, M.: The future sea-level contribution of the Greenland ice sheet: a multi-model ensemble study of ISMIP6, The Cryosphere, 14, 3071–3096, https://doi.org/10.5194/tc-14-3071-2020, 2020.

---

## Author Comment (AC4)

We thank Reviewer 2 for reviewing our work and for providing constructive comments and recommendations to improve our manuscript. We have worked on updating the manuscript to include the suggestions of Reviewer 2 as best as possible. We believe that the revisions have improved the quality of the manuscript, and are addressing the concerns raised by Reviewer 2. This document consists of a point-by-point reply to all the comments. In this document, the comments from Reviewer 2 are in blue, while our responses are in black. New text that has been included in the manuscript is in *black italic*.

A revised version of the manuscript will accompany this response. We will also upload a "tracked changes" version of the manuscript where all the modifications are highlighted. Line numbers in this document refer to line numbers of the revised manuscript without tracked changes.

General comment 1:
Accounting for uncertainty in ice-sheet modeling is of paramount importance and tools like the one presented here are important and worthy of publication. However, I find the exposition hard to follow. Details about the stochastic model are at times buried in the presentation of the numerical experiments and some important details are missing. In my understanding the StISSM provides two stochastic processes in time (potentially different in each sub-domain): an autoregressive process and an one based on Gaussian noise.

We agree entirely with this comment. We have added extensive emphasis on the difference between additive Gaussian white noise processes and stochastic autoregressive processes. In the manuscript, we consistently use the generic variable "y" to represent the former and the generic variable "eta" to represent the latter. We now specify the difference between both explicitly before introducing the equation for autoregressive processes (l 213-215):

*We use the notation $\eta$ to represent a variable governed by a stochastic AR process, in contrast to y, which is governed by an additive Gaussian white noise process (see Eq. (1)).*

In Sect. 2.3, we have also clarified that both y and eta are ISSM variables that are represented as random processes in StISSM v1.0 (l 239-240):

*It should be noted that $\eta$ (Eq. (7)) and y (Eq. (1)) are both the realization of a random process used as an ISSM variable, with the former being an autoregressive stochastic process and the latter an additive Gaussian white noise process.*

General comment 2:
It also allows to run ensemble members in parallel, although it is not clear how the computational resources (cores, memory) are used.

We have expanded Sect. 2.4 to provide more details about these aspects. Please note that most of the management of computational resources, of the output generation, of the debugging files, and of the run monitoring are identical to what is done in the standard ISSM model. The only difference is that different ensemble members are run separately (i.e., in parallel on different nodes). Because ensemble members do not need to share information with each other and the overhead associated with stochastic generation is negligible, the computational performance of StISSM v1.0 is effectively identical to ISSM as documented in prior studies (e.g., Larour et al. 2012). See l 254-261:

*The runs of the different members are executed on different nodes, and each separate member run can further be parallelized on different processors using the usual ISSM parallelization capabilities (Larour et al., 2012). StISSM v1.0 allows any of the possible output variables of ISSM to be saved, and at a user-specified frequency in order to manage output size. Output variables can be scalar (e.g., total ice mass) or multi-dimensional (e.g., ice thickness at each mesh element) fields. From the outputs, ensemble statistics can be computed (example codes are provided, see Sect. Code and data availability). Log files are automatically generated for debugging purposes, as is usual for ISSM runs. The implementation of ensemble simulations in StISSM v1.0 is straightforward, as illustrated in Algorithm 1.*

Furthermore, we now also explicitly state in Algorithm 1 that the user can specify the requested outputs and the temporal frequency at which outputs are saved.

General comment 3:
Statistical quantities (e.g. moments, p-values) are computed and reported in the Results section, but it is not clear whether these are computed directly by StISSM or how the data from the ensembles are collected.

Output from the ensemble runs are retrieved as is usual in the ISSM model (Larour et al., 2012). Statistical quantities can be computed from the outputs. We provide the scripts that we used to compute these quantities from the model outputs in the Zenodo dataset associated to this publication (see Code and Data availability section). This has been specified in l 254-261 (see text in the response to General comment 2 above).

General comment 4:
Finally, how is the stochastic layer implemented? How is it coupled to ISSM? Is it a driver to ISSM? Is it implemented in C++, Python or other languages?

All the StISSM v1.0 capabilities are implemented within the source code of ISSM itself. Using stochasticity or not is specified by the user when configuring their simulations. This is specified at the very start of our Methods section (l 104-106):
*The new stochastic capabilities are implemented within the core of the source code of ISSM. We refer readers to Larour et al. (2012) for a general description of ISSM. Usage of stochasticity is optional: if turned off, ISSM simulations are fully deterministic.*

To further address this concern from Reviewer 2, we specify in subsection 2.2 (l 199-201):
*All the stochastic schemes are implemented in the C++ source code of ISSM and are integral parts of the core of the model, but the schemes are not called if stochasticity is not required by the user.*

General comment 5:
A good part of the paper is devoted to using the StISSM for applying different stochastic parametrizations to two synthetic ice problems and a real ice sheet (Greenland). The numerical experiments are well thought out but I don't think it is particularly useful to target three different applications. I think it would have been better to target only one application (maybe a glacier) and show the effect of different choices of the parametrizations on the glacier evolution and mass balance. Targeting different (and complex) problems makes it harder to understand the impact of different parametrizations, without adding much in terms of explaining or demonstrating the stochastic model.

Our choice of model experiments is motivated by the fact that this publication is a model development study. Therefore, the main goal of the experiments is to demonstrate the range of capabilities of StISSM v1.0. We agree with the reviewer that there is a large scope for using StISSM v1.0 to addressing more specific science questions, in which case it would be relevant to target one system and do a comprehensive sensitivity study to various parameterizations.
Here, our three model experiments have specific purposes. We believe that they are all relevant to demonstrating the features of StISSM v1.0 and/or to demonstrating the importance of stochastic processes in idealized ice sheet model experiments. Thus, we prefer to preserve all three experiments in the main text as they provide the broadest possible demonstration of the capabilities of StISSM v1.0 which will be useful to the audience of Geoscientific Model Development.
MISMIP+: These experiments show the effect of different choices of autocorrelation timescales in the melt parameterization on the evolution and mass balance of an idealized glacier. We believe that this addresses the request mentioned by Reviewer 2.
IQIS: The stochastic SMB experiment (IQIS_1) demonstrates that StISSM v1.0 can use different subdomains with a prescribed correlation between them. This is an important tool of the model and it needs to be

demonstrated in a simple experiment. The stochastic calving experiment (IQIS_2) shows that even an idealized ice sheet configuration is subject to strong noise-induced drift. The stochastic calving and SMB experiment (IQIS_3) demonstrates that correlation can be prescribed between different variables, which is another important feature of StISSM v1.0. This experiment also demonstrates that there is no one-to-one correspondence between the ensemble spread of IQIS_3 and those of IQIS_1 and IQIS_2, despite IQIS_3 only combining the stochastic forcings of IQIS_1 and IQIS_2. Finally, IQIS_4 shows how the ensemble PDF is affected by temporal persistence in frontal ablation compared to white noise in frontal ablation (which is shown in IQIS_2).
GrIS: The purpose of this experiment is to show that StISSM v1.0 can be applied at the scale of a realistic ice sheet. We have used reasonably realistic climatic forcing fields for the purpose of demonstration.

General comment 6:
Moreover, given that the main novelty introduced by StISSM is that it provides parametrizations, I would have expected more emphasis on 1) why to choose a specific stochastic (e.g autoregressive) processes for modeling, e.g., the surface mass balance (SMB), 2) how the stochastic process compares, in a statistical sense, with available time-series of SMB,

We agree with Reviewer 2 that this is an important scientific question. However, the purpose of this study is to develop the numerical capabilities to exploit such knowledge in ice sheet model simulations. Constraining precisely the statistics of variability in climate and in glaciological processes is, in our opinion, beyond the scope of this work. However, we are working actively on these research questions. Two future publications are currently in preparation, and we are involved in cross-institution research projects to better quantify variability in climatic forcing and poorly constrained ice sheet processes. We agree with the reviewer that this important aspect should be more emphasized in the manuscript of StISSM v1.0. For this reason, we have added some information in the Discussion section (l 569-572):
*In this first version of a stochastic ISM, we have implemented simple forms of stochastic processes and statistical generators of climate forcing: additive Gaussian white noise and autoregressive time series models, respectively. This lays the groundwork for future, more sophisticated schemes specifically calibrated to represent the details of variability in glaciological and climatic processes. In particular, priorities are to implement seasonality in the statistical models, more complete time series models (e.g., autoregressive moving average, ARMA), and to allow for other forms of noise forcing in order to represent non-Gaussianity in components of the climate and ice sheet systems (e.g., Perron and Sura, 2013).*

Please also note that we reiterate the importance of constraining the parameterizations in the last paragraph of the Conclusion (l 640-643):
*In the future, calibration work will be needed to constrain the statistical models for climate forcing, as well as the variability in unresolved glaciological processes such as calving and hydrology. Such an effort will require combining observations, theory, and results from high-fidelity model experiments to understand the internal spatiotemporal variability of processes of interest.*

General comment 7:
(…) 3) what is the impact of using a first-order versus an higher-order autoregressive process, and so on.
We provide now a better physical intuition of how first-order autoregressive processes compare to higher-order ones (l 220-222):
*The order p of an AR process allows to capture multiple degrees of freedom influencing η, and that may act on different timescales (von Storch and Zwiers, 1999). Using a higher-order AR model thus allows to capture more complicated temporal variability, but implies the risk of overfitting if the calibration time series are too short.*

Additional comments:
Eq. (1). Is the Gaussian noise uniform in space even if the mean value is not? Please specify this in the text and discuss this choice. Please specify that the "mean" is intended in time, not in space (if I understand correctly).
We have clarified that the mean is intended in time (l 123):

*If y has a prescribed temporal mean value y^bar,*

We have also clarified upfront that the noise can vary spatially across the model domain, and refer the reader to more details in Section 2.2 (l 127-128):

*As explained in Sect. 2.2, $\sigma\_y$ is fixed in time but can be variable in space, hence allowing $\varepsilon\_y$ to vary across the model domain.*

line 153: would it make sense to have different stochastic time steps for different parameters?

This is a future development that can be considered. This was also pointed out by Reviewer 1. It is not straightforward to implement different stochastic time steps for correlated variables, because noise terms cannot be generated independently in this case. However, we have included this prospect in the manuscript (l 169-171):

*At this stage, StISSM v1.0 uses an identical stochastic time step for all variables modeled with additive Gaussian white noise (Eq. (1)), but implementing different stochastic time steps is a possible avenue for future development.*

line 164: OK, so the spatial stochasticity is introduced at the subdomain level. This should have been explained before, in the introduction and, in more detail in section 2.1 where it should be explained that eq. (1) is at the subdomain level.

See our response to the comment on Eq. (1) (first Additional comment).

eq. (7): I do not fully understand the purpose of the intercept and trend terms. Also, what is the choice for beta_0 and beta_1 in the numerical experiments in sections 3?

We agree with Reviewer 2 that the explanations on the intercept and trend terms in the previous version were insufficient. We have reformulated Eq. (7) to make it rigorous, and to make it clear how an autoregressive variable is computed. The purpose of the trend (beta_1) term is to allow for a deterministic linear trend in time (e.g., warming of ocean waters or decrease in SMB). The purpose of the intercept (beta_0) term is to allow for a non-zero baseline of the variable. In particular, if beta_1 is equal to zero, beta_0 represents the temporal mean of the variable. We have now specified the values of beta_0 and beta_1 in all the experiments introduced in Section 3.

line 235: can you detail how you manage resources (nodes, cores, memory) when you run in parallel multiple members of the ensemble (each of the members might need to be distributed on several ranks). Do you use any strategy to reduce I/O and storage when running large ensembles? Any strategy to monitor the runs (e.g. what happens if a few of the 500 simulations in the ensemble fail?)

Please see our response to the General comment 2.

eq (8) and (11): this is very minor, but I think that the use of squared terms "$C\_W^2$" and "$C\_B^2$" is poor notation. I know it is somewhat common, but it is misleading because it sort of implies that $C\_W$ and $C\_B$ have some physical meaning. Using the square to denote positive quantities (if that's the reason for the square) is hardly defensible because there are a lot of other physical variables (e.g. thickness) or coefficients (flow factor) that are positive (or nonnegative) and they are not denoted with a square of some other quantity. I would suggest dropping the square and using directly the coefficient $C\_W$ and $C\_B$.

We understand the point of view of Reviewer 2. We have changed all the terms $C\_W^2$ and $C\_B^2$ to $C\_W$ and $C\_B$, respectively.

Sections 3 and 4: The rigid separation of the "Model experiments" and "Results" sections makes it harder to follow the exposition. I think that the Results part should follow the Model Experiments part for each of the three examples.

We believe that this is a valid concern. We did consider this possibility, even before the first submission of the manuscript. However, we find it preferable to keep the "Model experiments" and "Results" sections separate. Some readers may be particularly interested in how StISSM v1.0 works, and how to set up experiments with the model. These readers can focus on the Model experiments section. In contrast, some other readers may be interested in understanding the consequences of stochastic forcings on idealized ice sheet model experiments,

without feeling the need to go through the details of the model experiment setup. These readers can focus on the Results section. Finally, we believe that our use of subsections makes it easy to go back-and-forth between the explanations on the model experiment setup and the corresponding results. For all these reasons, we have preferred to keep these two sections separate.

We thank you for your constructive comments.
Vincent Verjans, on behalf of all authors